# Finding General Equilibria in Many-Agent Economic Simulations using Deep Reinforcement Learning

## Abstract

Real economies can be seen as a sequential imperfect-information game with many heterogeneous, interacting strategic agents of various agent types, such as consumers, firms, and governments. Dynamic general equilibrium models are common economic tools to model the economic activity, interactions, and outcomes in such systems. However, existing analytical and computational methods struggle to find explicit equilibria when all agents are strategic and interact, while joint learning is unstable and challenging. Amongst others, a key reason is that the actions of one economic agent may change the reward function of another agent, e.g., a consumer's expendable income changes when firms change prices or governments change taxes. We show that multi-agent deep reinforcement learning (RL) can discover stable solutions that are $\epsilon$-Nash equilibria for a meta-game over agent types, in economic simulations with many agents, through the use of structured learning curricula and efficient GPU-only simulation and training. Conceptually, our approach is more flexible and does not need unrealistic assumptions, e.g., market clearing, that are commonly used for analytical tractability. Our GPU implementation enables training and analyzing economies with a large number of agents within reasonable time frames, e.g., training completes within a day. We demonstrate our approach in real-business-cycle models, a representative family of DGE models, with 100 worker-consumers, 10 firms, and a government who taxes and redistributes. We validate the learned meta-game $\epsilon$-Nash equilibria through approximate best-response analyses, show that RL policies align with economic intuitions, and that our approach is constructive, e.g., by explicitly learning a spectrum of meta-game $\epsilon$-Nash equilibria in open RBC models.

## 1 Introduction

Real-world economies can be seen as a general-sum sequential imperfect-information game with many heterogeneous strategic agents, including consumer-workers, firms, and governments. Dynamic general equilibrium models (DGE) describe the economic incentives, activity, interactions, and constraints of (a large collection of individual) agents to model and predict *macro-economic* outcomes, such as productivity, equality, and growth (Heer & Maussner, 2009). Following *micro-economic theory* (Mas-Colell et al., 1995), the behavior of each agent is driven by incentives ("rewards"), e.g., consumers maximize utility ("happiness"). For economic analysis, it is key to find the strategic equilibria. e.g., at an $\epsilon$-Nash equilibrium, no agent would unilaterally change its behavior, assuming, for example, that all agents act rationally and optimally (Nisan et al., 2007).[1]

Existing analytical and computational methods often struggle to find explicit DGE equilibria. First, the nonlinear structure of DGE models makes it challenging to find optimal agent policies explicitly. For example, an often-used analytical approach is to solve a linearized version of the DGE dynamics, akin to a Taylor expansion. However, a linearized DGE model may have fewer or different equilibria compared to the full DGE model, and solving the linearized model may choose one equilibrium arbitrarily when many are possible. Furthermore, they may only be good approximations around

---

[1]This could be expanded to more closely mimic human behavior through "boundedly rational" agents that may act suboptimally or whose affordances or objectives encode cognitive biases or limits (Kahneman, 2003).

steady-state equilibria and not be valid in the presence of shocks, e.g., when agents change their behavior dramatically in time (Boneva et al., 2016; Atolia et al., 2010). Second, formal solutions methods, e.g., backwards induction and related dynamic programming methods, often cannot be solved explicitly and only characterize optimal policies implicitly (Stokey et al., 1989). For example, analytical work in economics has studied implicit characterizations of the impact of taxes. One can derive analytical expressions for the distortion in consumer savings or capital due to a taxation policy or the asymptotic behavior of taxes in the far future, the tax rate on the highest-skill agent, or the structure of incentives (Golosov et al., 2011; Acemoglu et al., 2010). However, these methods generally cannot find the full tax policy explicitly or only describe part of it. Third, computational methods can suffer from the curse of dimensionality, e.g., when there is a large number of agents. In particular, agent incentives are often misaligned and pose a complex general-sum game. However, enumerating and selecting equilibria in general-sum games is an unsolved challenge (Bai et al., 2021).

Multi-agent deep reinforcement learning (RL) (Sutton & Barto, 2018) is a powerful framework to model agent behavior and has several key advantages for economic analysis. Deep neural network policies can learn nonlinear predictive patterns over high-dimensional state-action spaces, while deep value functions can model complex reward function landscapes over long time horizons. Notably, deep RL has achieved superhuman performance in high-dimensional, sequential games (Silver et al., 2017; Vinyals et al., 2019; OpenAI, 2018), suggesting it can learn (approximate) strategic equilibria in complex settings. Moreover, one does not have to approximate nonlinear world dynamics (e.g., as in the DGE model); RL has been successful in highly nonlinear environments (Tassa et al., 2018).

However, *deep RL is still not commonly used in economics*, although recent work has started to explore its use in economics (Zheng et al., 2020; Trott et al., 2021; Hill et al., 2021; Chen et al., 2021). In particular, deep MARL poses several challenges in the context of economic simulations. Our work is to our knowledge the first empirical application of multi-agent deep RL to general equilibrium economic simulation where all agent types learn.

First, a general MARL challenge is that each agent faces a non-stationary environment *when multiple agents learn simultaneously*; each agent's experienced reward may depend on what other agents do. For example, a consumer may not be able to buy a desired good if there is not enough supply generated by the corresponding firm. Second, economic agents typically have private information that is not available to other agents, e.g., a worker's skill level and behavioral policy (parameters) are not observable by firms. This means that learning has to be (partially) decentralized, as in Dec-POMDPs (Oliehoek & Amato, 2016). Moreover, a key feature of economic models is that *the actions of one economic agent may change the shape of the reward function* of another agent, e.g., a consumer's expendable income changes when firms change prices or governments change taxes. Hence, agents change the optimization problem for other agents during training.

In MARL, especially deep MARL, joint learning can be unstable, and naive approaches are challenging to get to work. In the context of our economic models, individually trained policies easily get stuck in trivial equilibria where no labor or production occurs.

**Contributions** To address these challenges, we develop MARL techniques that enable finding equilibria in DGE models with many strategic agents. We analyze DGE models with **hundreds of heterogeneous RL agents**, a scale that has not been studied before with strategic agents. Given the large number of agents in the simulation, we run both simulation and RL on a GPU, following the WarpDrive framework (Lan et al., 2021), so training converges in hours (rather than days). WarpDrive uses deep RL to train neural network policy models that are stored on the GPU, using simulation data that is generated on that same GPU. This system design accelerates training significantly by avoiding copying data unnecessarily and placing both simulation and agent models on the same device.

From an **algorithmic** perspective, we generalize the curriculum approach from Zheng et al. (2020) to structured learning curricula for simulations with multiple agent types. This approach yields non-trivial solutions *more stably* compared to using agent-independent RL. We also show our approach is *sound*, e.g., by showing that an RL government can improve social welfare vs fixed baselines.

From a **modeling** perspective, our approach is more flexible and can find stable solutions in a variation of real-business-cycle (RBC) models (Pierre Danthine & Donaldson, 1993), a family of DGE models. We find equilibria in closed and open RBC economies, the latter has a price-taking export market.

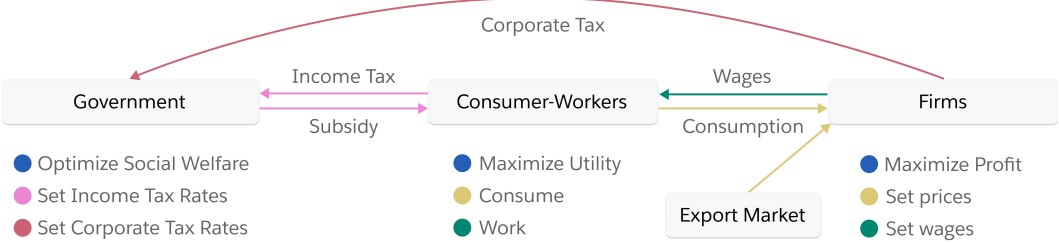

Figure 1: **RBC model with consumers, firms, and governments.** Arrows represent money flow. Consumer-workers earn wages through work and consume goods from firms. They also strategically choose *which* firm to work for and *which basket of goods* to buy, but this is not explicitly visualized. Firms produce goods, pay wages, and set a price for their goods. They also invest a fixed fraction of profits to increase capital. The government taxes labor income and firm profits, and redistribute the tax revenue through subsidies to the consumer-workers. Firms can also sell goods to an external export market, which acts as a price-taker that is willing to consume goods at any price.

From a **game theory** perspective, we show that our MARL approach can explictly find $\epsilon$-Nash equilibria for the meta-game over agent types, without approximating the DGE model. Here, the meta-game equilibrium is a set of agent policies such that no *agent type* can unilaterally improve its reward by more than $\epsilon$ (its *best-response*). Previous solutions only could do so implicitly or for approximations of the DGE dynamics. Furthermore, we learn a *spectrum of equilibria* in open RBC economies, i.e., explicitly learn the equilibrium behavioral policies.

More general types of equilibria may exist in DGE models but can be hard to validate. For instance, there may be asymmetric Stackelberg-like equilibria, where a leader (e.g., the government) acts first (e.g., sets taxes), and the followers (e.g., consumers and firms) respond. This differs from symmetric Nash equilibria where all agents act simultaneously. However, analyzing the equilibria in Stackelberg-like, general-sum games is an open challenge (Bai et al., 2021). For instance, with 2 or more (heterogeneous) followers, finding the Stackelberg best-response to a fixed leader requires finding multi-agent equilibria for the followers. This is computationally expensive and there is no known provably convergent algorithm for multi-level Stackelberg games. Instead, we use MARL to converge to a stable solution and analyze best-responses to evaluate to what extent it is an "equilibrium".

Moreover, *which* equilibrium agents converge to ("equilibrium selection") may depend on, e.g., the 1) world dynamics, 2) initial conditions and policies, 3) learning algorithm, and 4) policy model class. Our framework allows us to study how these factors relate to which equilibria can be found.

## 2 RELATED WORK

We highlight some related work; for a more thorough discussion, see Appendix A. The AI Economist uses RL to design optimal economic policies in simulation environments (Zheng et al., 2020; Trott et al., 2021). Other works also use deep RL for economic modeling, but they typically only allow one agent type to learn, and make assumptions like market-clearing (Hill et al., 2021; Chen et al., 2021). Perhaps most similar to our work is the agent-based model of Sinitskaya & Tesfatsion (2015), which uses tabular Q-learning in a discretized environment for small numbers of identical consumer-workers and firms (but no government), with looser market-clearing constraints. Discussed in the Appendix are other works related to the use of RL for games with leader-follower or Stackelberg structure, and the use of computational methods for approximating equilibria.

## 3 CASE STUDY: REAL-BUSINESS-CYCLE MODELS

We show how RL can find equilibria in real-business-cycle models (RBC), a representative DGE model in which consumers earn income from labor and buy goods, firms produce goods using capital and labor, and the government taxes income and profits (Pierre Danthine & Donaldson, 1993), see Figure 1. RBC models are stylized and may not fully describe reality (Summers et al., 1986). However, RBC models are a suitable environment to validate the use of RL, as they feature

heterogenous agents with nonlinear economic interactions, making it challenging to find equilibria. Below, we describe the RBC dynamics; for low-level details, see Appendix D. At a high level, our model includes worker-consumers, price-and-wage setting firms, and a government who sets tax rates and redistributes. A key point about our model is that we do not assume as part of the environment that prices and wages are set so that markets clear at each time step – that is, goods may be overdemanded and firms are free to set prices and wages lower or higher than would balance supply and demand. These assumptions are an essential part of the techniques used to derive analytic solutions – avoiding this modeling choice requires using other techniques.

**Agent Types.** Formally, our RBC model can be seen as a Markov Game (Littman, 1994) with multiple agent types and partial observability. The simulations proceeds in finite-length episodes with $T$ timesteps. Each timestep represents a quarter of 3 months. At each timestep $t$, we simulate

- firms and their goods who use the labor of the worker-consumers to each produce a different good, indexed by $i \in I$,
- consumers, indexed by $j \in J$, who work and consume goods, and
- a government who sets a tax rate on income from labor and on revenue from selling goods.

Each agent iteratively receives an observation $o_{i,t}$ of the world state $s_t$, executes an action $a_{i,t}$ sampled from its policy $\pi_i$, and receives a reward $r_{i,t}$. The environment *state* $s$ consists formally of all agent states and a general world state (see Appendix D for details). At each timestep $t$, all agent simultaneously execute actions. However, some actions only apply to the next timestep $t + 1$. For example, the government sets taxes *that will apply at the next timestep* $t + 1$. These tax rates are observable by the firms and consumer-workers at timestep $t$; hence, they can condition their behavioral policy in response to the government policy. Similarly, at timestep $t$, firms set prices and wages that will be part of the global state at the next $t + 1$. This setup is akin to, though not exactly corresponding to, a Stackelberg leader-follower structure, where, for example, the government (leader) moves first, and the firms and consumer (followers) and move second (von Stackelberg et al., 2010). This typically gives a strategic advantage to the followers: they have more information to condition their policy on.

**Consumer-Workers.** Individual people both consume and work; we will refer to them as *consumer-workers*. At each timestep $t$, person $j$ works $l_{j,t}$ hours and consumes $c_{i,j,t}$ units of good $i$. Each person chooses to work for a specific firm $i$ at each timestep. Consumer-workers also (attempt to) consume $\hat{c}_{i,j,t}$. Each firm's good has a price $p_i$ (set by the firms, described below), and the government also sets an income tax rate. However, consumers cannot borrow or outspend their budget: if the cost of attempted consumption exceeds the budget, then we scale consumption so that $\sum_i p_i \hat{c}_{t,i,j} = B_j$.

Moreover, the realized consumption depends on the available inventory of goods ($y_{i,t}$, described below). The total demand for good $i$ is $\hat{c}_{i,t} = \sum_j \hat{c}_{i,j,t}$. If there is not enough supply, we ration goods proportionally:

$$c_{i,j,t} = \min\left(1, \frac{y_{i,t}}{\hat{c}_{i,t}}\right) \hat{c}_{i,j,t}. \tag{1}$$

Consuming and working change a consumer's budget $B_{j,t}$. Consumer $j$ has labor income $z_{j,t} = \sum_i l_{i,j,t} w_{i,t}$; each firm pays a wage $w_{i,t}$. The cost of consumption is $\sum_i p_{i,t} \cdot c_{i,j,t}$ Moreover, with tax rate $\tau_t$, workers pay income tax $\tau_t \cdot z_{j,t}$; the total tax revenue $R_t$ (which also includes taxes on the firms, described below) is redistributed evenly back to workers. In all, consumer budgets change as:

$$B_{t+1,j} = B_{j,t} + (1 - \tau_t)z_{j,t} + \frac{R_t}{|J|} - \sum_i p_{i,t} \cdot c_{i,j,t}. \tag{2}$$

Each consumer optimizes its behavioral policy to maximize utility:

$$\max_{\pi_j} \mathbb{E}_{c,l \sim \pi_j}\left[\sum_t \gamma_c^t \sum_i u(c_{i,j,t}, l_{i,j,t}, \theta_j)\right], \quad u(c, l, \theta) = \frac{(c + 1)^{1-\eta} - 1}{1 - \eta} - \frac{\theta}{2}l_t, \tag{3}$$

where $\gamma_c$ is the consumer's discount factor. The utility function is a sum of isoelastic utility over consumption and a linear disutility of work with coefficient $\theta_j$ that can vary between workers.

**Firms.** At each timestep $t$, a firm receives labor from workers, produces goods, sells goods, and may invest in capital. At each timestep $t$, it sets a price $p_{t+1,i}$ for its good and chooses a wage $w_{t+1,i}$ to pay, both *effective at the next timestep $t+1$*. If a firm invests $\Delta k_{i,t}$ in capital, its capital increases as $k_{i,t+1} = k_{i,t} + \Delta k_{i,t}$. Using its capital $k_{i,t}$ and total labor $L_{i,t} = \sum_j l_{i,j,t}$ (hours worked), a firm $i$ produces $Y$ units of good $i$, modeled using the *production function*:

$$Y_{i,t} = A_i k_{i,t}^{1-\alpha} L_{i,t}^{\alpha}, \tag{4}$$

where $0 \leq \alpha \leq 1$ sets the importance of capital relative to labor. Also, consumers buy $C_{i,t} = \sum_j c_{i,j,t}$ units of good $i$. Accordingly, inventories change as $y_{t+1,i} = y_{i,t} + Y_{i,t} - C_{i,t}$. Inventories are always positive, as only actually produced goods can be consumed. The firms receive a profit (or loss) $P$, pay taxes on their profit, and experience a change in their budget $B$:

$$P_{i,t} = p_{i,t} C_{i,t} - w_{i,t} L_{i,t} - \Delta k_{i,t}, \; B_{t+1,i} = B_{i,t} + (1 - \sigma_t) P_{i,t}, \tag{5}$$

where $\sigma$ is the corporate tax rate. The government receives $\sigma_t P_{i,t}$. Firms may borrow and temporarily be in debt (negative budget), but should have non-negative budget at the end of an episode (*no-Ponzi condition*). This may allow firms to invest more, which may lead to higher future economic growth. Each firm optimizes its behavioral policy to maximize profits as in Equation 5:

$$\max_{\pi_i} \mathbb{E}_{p,w,\Delta k \sim \pi_i} \left[ \sum_t \gamma_f^t P\left(p_{j,t}, w_{j,t}, \Delta k_{j,t}\right) \right], \tag{6}$$

where $\gamma_f$ is the firm's discount factor. The firm gets a negative penalty if it violates *no-Ponzi*.

**Government.** The government, or social planner, indexed by $p$, sets corporate and income tax rates, and receives total tax revenue $R_t = \sigma_t \sum_j z_{j,t} + \tau_t \sum_i P_{i,t}$. As a modeling choice, this revenue is redistributed evenly across consumer-workers; as such, the government's budget is always 0. The government optimizes its policy $\pi_p$ to maximize social welfare:

$$\max_{\pi_p} \mathbb{E}_{\tau,\sigma \sim \pi_p} \left[ \sum_t \gamma_p^t \mathrm{swf}\left(\tau_t, \sigma_t, \boldsymbol{s}_t\right) \right], \tag{7}$$

where $\mathrm{swf}\left(\boldsymbol{s}_t\right)$ is the social welfare at timestep $t$, and $\gamma_p$ is the government's discount factor. In this work, we consider two definitions of social welfare (although many other definitions are possible): 1) *total consumer utility* (after redistribution), or 2) *total consumer utility and total firm profit*.

**Open and Closed Economies via Export Markets.** We consider both open and closed economies. In the open economy, firms can also sell goods to an external market which acts as a *price-taker*: their demand does not depend on the price of a good. Operationally, export happens after worker-consumers consume. The export market has a minimum price $p_{\mathrm{export}}$ and a cap $q_{\mathrm{export}}$. If the price of good $i$ is greater than the minimum price ($p_{i,t} > p_{\mathrm{export}}$) then the additional export consumption is $c_{t,\mathrm{export}} = \min(q_{\mathrm{export}}, y_{i,t} - C_{i,t})$, at price $p_{i,t}$, i.e., the exported quantity is insensitive to the price.

From a learning perspective, the export market prevents firms from seeing extremely low total demand for their good, e.g., when prices are exorbitantly high and consumers do not want or cannot consume the good. In such cases, an on-policy learner that represents a firm may get stuck in a suboptimal solution with extremely high prices and no production as consumers cease to consume in response.

## 4 Multi-Agent RL and Structured Curricula

A key idea of our approach is the use of structured multi-agent curricula to stabilize learning, where each individual agent uses standard RL. These curricula consist of staged training, annealing of allowed actions, and annealing of penalty coefficients, see Figure 2.

**Base RL.** Firms and governments use 3-layer fully-connected neural network policies $\pi\left(\boldsymbol{a}|\boldsymbol{s}\right)$, each layer using 128-dim features, that map states to action distributions. Consumer policies are similar, using separate heads for each action type, i.e., the joint action distribution is factorized and depends on a shared neural network feature $\varphi_t(s_t)$: $\pi(a_1, a_2, \ldots | s) = \pi(a_1 | \varphi, s)\pi(a_2 | \varphi, s) \ldots$ (omitting $t$ and $s_t$ for clarity). Any correlation between actions is modeled implicitly through $\varphi_t$. There is a

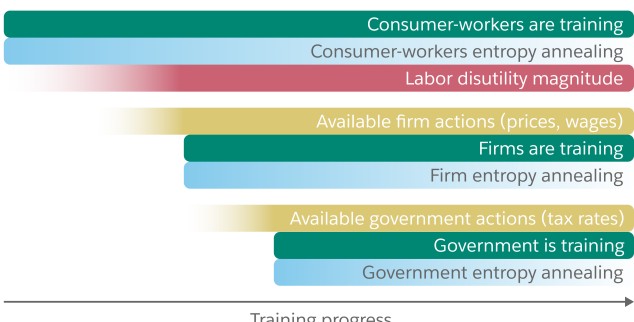

Figure 2: **Structured learning curricula.** Colored bars show how annealing activates over time. Consumers are always training, use a decaying entropy regularization, and increasingly experience their labor disutility. Firms start with fixed prices and wages, and are slowly allowed to change these. Once the full price and wage range is available, firms start to train. Similarly, tax rates start at zero and are slowly allowed to change. Once the full tax range is available, the government starts to train.

single policy network for each agent type, shared across the many agents of that type. To distinguish between agents when selecting actions, agent-specific features (parameters like the disutility of work, production parameters, and for firms, simply a one-hot representation of the firm) are included as part of the policy input state. Thus, despite a shared policy for each agent type, we model some degree of heterogeneity among agents. We also learn a value function $V(\varphi_t)$ for variance reduction purposes. We compare policies trained using policy gradients (Williams, 1992) or PPO Schulman et al. (2017). See Appendix E for further training details. Both simulation and RL ran on a GPU with WarpDrive (Lan et al., 2021), see Appendix G.

**Intuition for Structured Curricula.** We define curricula based on these observations about our multi-agent learning problem: 1) during exploration, many actions may reduce utility, while few actions increase utility, 2) high prices or tax rates can eliminate all gains in utility, even though the consumer-worker did not change its policy, 3) for stable learning, agents should not adapt their policy too quickly when experiencing large negative (changes in) utility, and 4) in a Stackelberg game, the followers (e.g., consumers, firms) should get enough time to learn their best response to the leader's policy (e.g., the government). We now operationalize these intuitions below.

**Staged Learning and Action Space Annealing.** All policies are randomly initialized. We first allow consumers to train, without updating the policies of other agents. Initially, firm and government actions are completely fixed; prices and wages start at non-zero levels. We then anneal the range of firm actions without training the randomly initialized policy. This allows consumers to learn to best respond to the full scope of prices and wages, without firms strategically responding. Once firm action annealing is complete, we allow the firm to train jointly with the consumers. We then perform the same process, gradually allowing the government to increase its corporate and income tax rates, so that firms and consumers can react to a wide range of tax rates. Once the annealing process is complete, we allow the government to train to maximize welfare.

**Penalty Coefficient Annealing.** In addition to the action annealing, we anneal two penalty co-efficients. First, we slowly increase the consumers' disutility of work over time, which avoids disincentivizing work early in the training process. Second, as each agent starts training, we assign a high value (0.5) for the entropy coefficient in their policy gradient loss, and gradually anneal it down over time to a minimum of 0.1. This ensures that when the firm or government policies start training, their "opponent" policies are able to learn against them without being too quickly exploited.

**Many local equilibria** We expect that there are many possible equilibria in our game that our approach may converge to. Moreover, convergence may be only local, since we are not globally optimizing our policy parameters. Our training curriculum is designed to avoid trivial equilibria where the economy shuts down, but it may also bias which non-trivial equilibria are reached. However, we

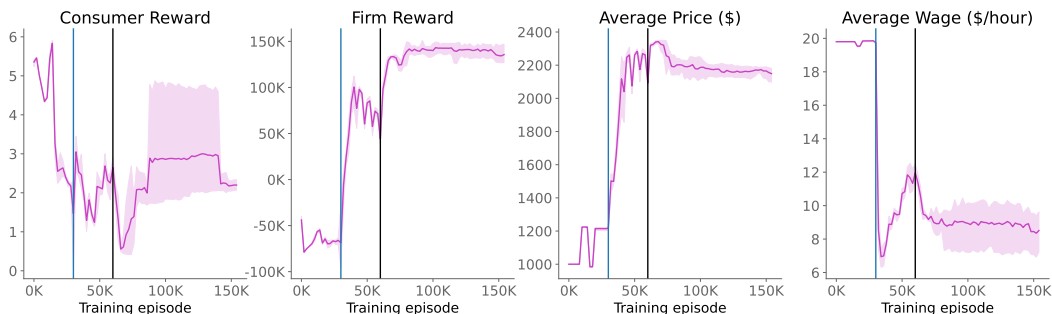

Figure 3: **Structured Curricula and Training, Open Economy, 3 runs.** In each plot, the blue (black) vertical line shows when the firms (government) start training, see Figure 2. **Left two plots:** Consumer and firm rewards during training. All runs converged to an approximate equilibrium, as confirmed by a best-response analysis. The outcomes were chosen to be qualitatively similar across runs. Once firms start training, their reward significantly increases. When the government starts training, firms get even higher reward, as the social welfare definition includes the firm's objective (profits). **Right two plots:** Average wages and prices (over firms) during training. Firms increase prices rapidly and lower wages once they start training.

| Agent | Consumer | Firm | Government |
|---|---|---|---|
| Reward improvement under best response | $< 3\%$ | $< 10\%$ | $< 1\%$ |

Table 1: Reward improvement under best response at the end of training as a fraction of the reward improvement during training, over 10 random seeds. We observed that besides a few anomalies, the improvements were in fact less than 0.2% for consumers, 5% for firms, and 0.1% for the government.

observe a spectrum of different outcomes, so we see our approach as a way to explore more equilibria than are possible with simplified and linearized models.

## 5    LEARNING RBC EQUILIBRIA AND METHOD VALIDATION

We show that our approach is sound and can explicitly find spectra of approximate RBC equilibria. We study variations of our RBC model with 100 consumers and 10 firms. For all simulation parameter settings, see Table 2 and Appendix D. We repeat all experiments with 3 random seeds.

**Structured Curricula during Training.**    Figure 3 shows a representative result of training using our structured curricula. Although all RL agents aim to improve their objective, some agents may see it decrease as the system converges towards an equilibrium, e.g., the consumers in Figure 3. Empirically, we found that training collapses to trivial solutions much more often without curricula.

**Best Responses and Local Equilibrium Analysis**    We abstract the RBC as a normal-form meta-game between three players representing the agent types, i.e., the consumers, firms, and government. We test whether a set of learned policies is an $\epsilon$-Nash equilibrium for the meta-game by evaluating whether or not they are approximate best responses. Recall that agents of the same type share policy weights (e.g., consumers), but use agent-specific inputs to their policy, To find an approximate best-response, we train each agent type separately for a significant number of RL iterations, holding other agent types fixed, and see if this improves their reward by more than $\epsilon$. If so, the policies are not an equilibrium; if not, this suggests we found at least a *local* equilibrium. In general, we find empirically that best-responses improve rewards much more at the start compared to the end of training. (See Table 1.) Throughout, we report results where the policies have found an approximate equilibrium at the end of training. Note that the meta-game over agent types is different from the game over individual agents. In particular, a meta-game best-response (where *all* agents of the same type can best-respond) may not be an individual's best-response. A meta-game best-response may feature competitive-coordination dynamics between agents of the same type, e.g., consumers, such that their collective behavior decreases the reward for their agent type. However, this may feature adverse

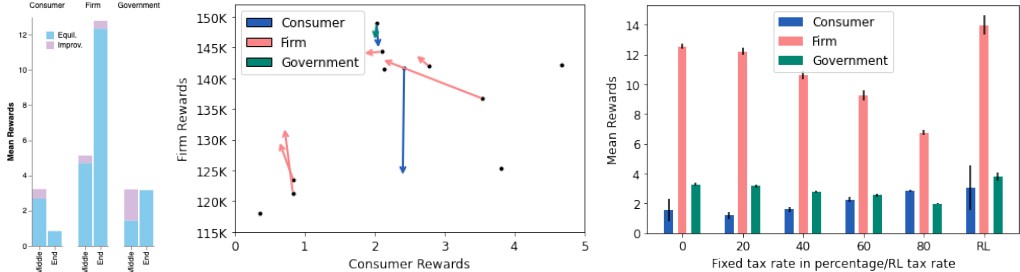

Figure 4: In each plot, the agent types (consumer, firm, and government, resp.) refer to cases when only that agent type is training. **Left: Best-Responses during Training.** The stacked bar chart shows the improvement in the mean rewards of an agent type. For firms and governments, the mean rewards are measured in units of $10^4$ and $10^3$, respectively. We compare the best response improvement in the middle and at the end of training. The improvement from best response is significant in the middle and much less at the end, indicating that training is closer to an equilibrium at the end. **Middle: Outcomes under Best-Response.** After training, typically neither consumer nor firm rewards change significantly under a best-response analysis. This holds generally for the approximate equilibria reported in this work. Here, rewards are on an absolute scale. In the figure, we display only those which change by more than 1%. **Right: Comparing Fixed and RL Government Policies.** Mean rewards for all agent types under fixed tax rates. Again, for firms and governments, the mean rewards are measured in units of $10^4$ and $10^3$, respectively. The mean rewards for the consumers increase with tax rates, whereas for the firms they decrease. An RL government can improve the mean reward for both types by appropriately adapting tax rates. RL policies increase the social welfare by almost 15% over the best baseline policy with a fixed tax rate.

behaviors for some individual agents, e.g., there may be free-riders that benefit from the collective performance while not putting in effort themselves. As we consider settings with a large number of agents (e.g., 100 consumers), we limit our analysis to meta-game approximate best-responses.

**Comparing with Baseline Government Policies.** To show that our approach is sound, we show that RL tax policies lead to improved social welfare (here, a weighted sum of firm and consumer rewards, see appendix C) compared with several different manually defined fixed tax rate policies. As observed in Figure 5, RL policies generate a social welfare ranging from 3000 to 4000, depending on the equilibrium (as noted above, multiple equilibria could be reached). Compared to that, the best social welfare achieved using fixed tax rates was 3160. Figure 4 shows the social welfare achieved under different various fixed tax rates. We note that social welfare improves by almost 15% for the best equilibrium under RL tax policy over the best equilibrium with fixed tax rates. This shows that the RL policy can adjust taxes across different rounds to improve average social welfare.

**Exploring RBC Equilbria.** Figure 5 visualizes and describes the learned RBC equilibria in open and closed RBC economies (with and without an export market). In both cases, we find multiple equilibria, although learning mostly converges to low-welfare equilibria in the closed economy. The open RBC model admits a wide spectrum of approximate equilibria with distinct outcomes (e.g., consumer utility, hours worked, prices, and taxes), and trends that align with economic intuitions. To study the differences between equilibria, Figure 6 shows a detailed comparison between two rollouts at two different equilibria. This reveals distinct equilibrium strategies, e.g., firms can profit by either focusing on consumers or the export market.

## 6 DISCUSSION

Daskalakis et al. (2009) showed that computing equilibria for general-sum games is hard in terms of computational complexity, even for simple matrix games. There is no theoretical guarantee that our framework can find *all* equilibria in sequential economic games. However, our best-response analysis suggests our framework does discover local equilibria. Hence, our current framework is at least

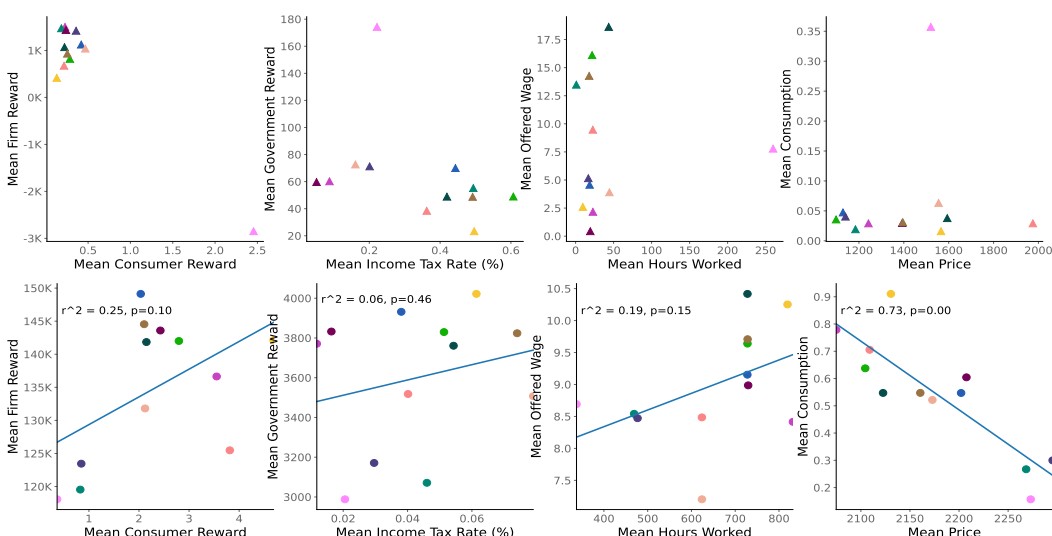

Figure 5: **Learned Equilibria in Open (Bottom) and Closed (Top) RBC Models.** Outcomes at convergence for the same experiments in Figure 4. Each point represents an approximate equilibrium, verified by an approximate best-response analysis. Points of the same color and shape correspond to the same run. **Top:** In the closed economy, training often converges to equilibria with low consumer reward and little production. In particular, social welfare (government reward) does not increase with higher tax rates, average labor does not change with wages, and consumption is unchanged with price. An exception is an equilibrium with significantly higher social welfare, labor, and consumption. This suggests multiple equilibria do exist, but non-trivial equilibria are harder to learn in the closed economy. **Bottom:** We learn multiple distinct, non-trivial equilibria in an open economy. Blue lines show linear regressions to the data. Consumer and firm rewards are positively correlated ($r^2 = 0.25$), e.g., if consumers earn more, they can consume more, yielding higher profits. Higher prices decrease mean consumption ($r^2 = 0.73$), lower wages decrease mean hours worked ($r^2 = 0.19$), and there is no strong signal that higher taxes improve social welfare ($r^2 = 0.06$).

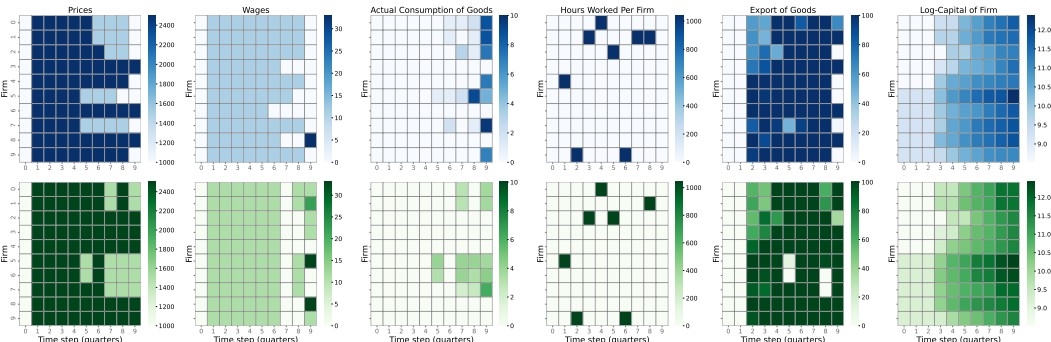

Figure 6: **Open Economy Rollouts, Two Different Equilibria.** We show actions and states for two representative runs at two qualitatively different equilibria. Heatmaps for the same metric use the same range of intensities, e.g., for prices in the top and bottom row. We observe that firms have different strategies: some set prices high and rely on exporting goods (for example, firm 3); others set prices lower and also sell to consumers (for example, firm 0). Consumers respond sensibly, only consuming when prices are low and only working when wages are not 0. The differences in initial firm endowments as well as evolution during the episode lead to different final capital levels.

an exploratory tool for finding qualitatively different equilibria, e.g., by varying initial conditions, environment parameters, or conditioning sampling, and provides exciting avenues for future research.

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

## A    RELATED WORK

**Computational Approaches to Find Equilibria.** Leibo et al. (2017) studied meta-game equilbria in sequential social dilemmas, showing that RL policies can be categorized into high-level strategy types akin to cooperation or defection in the classic social dilemma. Counterfactual regret minimization has yielded superhuman poker bots (Brown & Sandholm, 2018), an imperfect-information game; however, it can be computationally inefficient. Empirical game-theoretic analysis studies equilibria through agent simulations (Wellman, 2006), but is limited to games with interchangable agents with identical affordances (Tuyls et al., 2018) and do not easily scale to settings with heterogeneous agents.

Higher-order gradients (Foerster et al., 2017) and first-order gradient adjustments (Balduzzi et al., 2018) have sought to promote convergence to non-trivial equilibria. However, these methods make strong assumptions on what agents know, e.g., that agents can see the policy weights of other agents and/or know the full reward function, and may deform the original equilibria.

**RL for Economic Simulation.**    Agent-based modeling (ABM) studies economic simulations with individual agents. Early work mostly used rule-based agents or RL in finite state ABMs to study emergent phenomena (Bonabeau, 2002). A recent economics survey of ABMs (Haldane & Turrell, 2019) outlined the potential for RL agents, following its success in ML.

Several recent works have explored this idea. The AI Economist used RL to design optimal taxes to improve social welfare in high-dimensional simulations, where both agents and governments use RL policies (Zheng et al., 2020). RL also yields interpretable economic and public health policies in pandemic simulations (Trott et al., 2021). Danassis et al. (2021) takes a similar approach: they explore a multi-agent environment with harvesters participating in a common fishery, and a centralized price setter, and demonstrate that the price setter can outperform prices set by a market equilibrium.

Other contemporary work using deep RL includes Chen et al. (2021), which studies monetary policy with a single representative household RL agent, and Hill et al. (2021), which learns the value function of consumers in RBC models. These works are more limited, compared to ours. First, they use RL for one agent type, while the other agents (e.g., firms) use simple and fixed policies. Second, they assume markets always clear at each time step, i.e., prices are set to ensure supply and demand are balanced. However, this is an unrealistic assumption and causes slow simulations, requiring solving a nonlinear optimization problem at each timestep. Perhaps most similar to our work is Sinitskaya & Tesfatsion (2015), which deals with an economy with non-heterogeneous consumer-workers and price- and wage-setting firms (but no government). They to some extent enforce market clearing, by matching supply and demand for labor and goods in a double auction, although they do not enforce via constraints that there should be enough labor to produce demanded goods. Small numbers of consumers and firms are allowed to learn using tabular Q-learning. Even in their setting, they also observe that learning dynamics can collapse to trivial equilibria with no production or consumption.

In our work, we do not enforce market clearing constraints at all, simply rationing goods if there are not enough. Moreover, consumers have different parameters, and each firm has a different production function and produces a distinct good.

**RL and Stackelberg Games.**    In a Nash equilibrium, every player is playing a best response against every other player, so that no player can improve their outcome by unilaterally deviating. By contrast, in a Stackelberg game, one player is a leader who has the opportunity to move first; followers observe that move and choose their best response. As discussed in Zheng et al. (2020), there are aspects of Stackelberg games in economic simulations: a government who sets taxes or a firm setting wages and prices must make a choice anticipating a response by other agents. One popular setting for Stackelberg equilibria is in security games (An et al., 2017). In such games, the leader represents a defender who chooses where to deploy patrols; the attacker wishes to evade the patrols and knows the strategy chosen by the leader. Several works have studied the use of RL techniques in such settings (Wang et al., 2019; Trejo et al., 2016; Kamra et al., 2018; Bai et al., 2021).

**Theoretical guarantees of convergence**    Guarantees of convergence for model-free policy optimization are difficult to come by. Some theorems only apply to tabular policies (as in Srinivasan et al. (2018), and where convergence to Nash is not even guaranteed due to a non-sublinear regret bound). Other cases also apply only to very limited families of games, as in Zhang et al. (2019), which deals only with zero-sum linear-quadratic games. We deal with a general sum game with large numbers of agents and with neural network policies – all three of these properties mean that theoretical convergence guarantees are beyond the reach of theory.

## B    REPRODUCIBILITY STATEMENT

We discuss specific training details for all experiments such as hyperparameters in the Appendix. Upon acceptance, all code for this project will be made open source and publicly available for reproducibility purposes and further research. Upon request, we will share our code with reviewers and area chairs for review.

## C    ETHICS STATEMENT

Our work proposes a framework to model economies using Multi-Agent Reinforcement Learning and thus may be used to draw implications about the real world. Our findings and used simulations are purely for research purposes and should not be used to make decisions in real-world systems. Furthermore, our framework should not be used to explore methods to increase discrimination or unfairness in real-world systems.

**Assumptions, limitations, and ethical implications of using ML for economics.**    All choices in the economic simulation model, RL algorithms, reward functions, etc, play an important but difficult-to-understand role in equilibria selection and policy design. As in all ML applications, there are assumptions and limitations in the methodology. This has ethical implications for their use in future policy design applications.

**Mitigation strategies and interdisciplinary research.**    Economic simulation enables studying a wide range of economic incentives and their consequences, including models of stakeholder capitalism. However, the version of the simulation as used in this work is not an actual tool that should be used for policy making.

Many design choices influence the eventual policy recommendations. For example, the designer is free to set the social welfare objective that the government optimizes for. As such, it is crucial that these choices are debated and made in a socially acceptable fashion by all stakeholders, and made transparent and accessible to all.

More generally, to mitigate ethical risk, further mitigation strategies may include performing a what-if analysis over worst-outcomes, opening research results to domain experts (social scientists, ethics experts, etc), and open-sourcing the research results, amongst others. In all, the design and use of ML for policy recommendations will require robust, multilateral discussion and careful consideration of ethical risk, potential harm, and which trade-offs are being made.

We now detail some assumptions, limitations, and potential ethical risk among different dimensions of using ML for economics. *We stress that there can be more (unknown) aspects that we do not address here.* As such, we see this discussion as a starting point of discussion for the ML and economics community.

**Economic simulation and data.**    While the current version of the economic simulation provides only a limited representation of the real world, we recognize that future, large-scale iterations can still contain biases and unrealistic assumptions. Furthermore, non-representative simulation environments may result in biased policy recommendations. For instance, the under-representation of communities and segments of the work-force in training data might lead to bias in simulations that build on those and lead to biased AI policies. As such, collecting more representative data is a key challenge for future research in using ML for economic policy recommendations.

Our RBC model is a stylized model of real economies. RBC models are a commonly used class of economic models (see e.g. Smets & Wouters (2007)). However, as any model, it contains assumptions and stylizations. Future simulations may miss (un)known features that pertain, e.g., to equity and equality in the economy. Therefore, using simulations that are not representative or well-calibrated, can exacerbate or create new socio-economic issues.

We list a few salient features and assumptions below, although we cannot exhaustively enumerate all features that may be relevant in future research.

- Our RBC model features consumers that differ in skill and perform different amounts of work. However, we do not model more fine-grained distributional features, such as educational attainment, wealth, inheritance, geography, or others.

- Similarly, firms produce a single good only and can invest and pay wages. Any worker can work for any firm. We do not model hiring practices, the geographic location of firms, non-monetary incentives or benefits (e.g., health insurance). To accurately model inequity in the real world, including such features may be necessary.

- On the government and societal level, we model tax policies and simple redistribution of tax revenue. We do not model targeted redistribution, tax credits, application-specific subsidies (e.g., education support). We do not model trading, inflation, debt, and other macroeconomic features that may impact social groups disparately.

Our RBC is more general than commonly used models: we do not enforce market clearing, for instance. Market clearing is an unrealistic assumption that supply always meets demand. Economic theory uses such constraints to make analysis tractable. In contrast, our learning approach is flexible and does not require such simplifying assumptions. We also assume that all agents can observe the wages offered by all the firms. However, in the real world not all agents have equal access to information – and this is a feature that can be studied by future research. As such, we view the flexibility of our learning approach as a positive, in that our framework may allow for studying more representative models.

**Choice of economic incentives and rewards.** Agents optimize their behavior given economic incentives, as modeled by their reward function. As such, future economic AI policies should clearly describe for which reward function they were optimized. Furthermore, more research is needed to understand how the choice of reward function influences the resulting policies, and how social and ethical values can be transparently encoded in reward functions. It is also an open question which ethical/social norms and values can or cannot be quantified, and how to encode trade-offs between conflicting values.

For example, the planner optimizes its policy to maximize "social welfare", a standard economics concept. However, the definition of social welfare heavily influences the resulting policy and social outcomes. For example, Zheng et al. (2020) used equality times productivity as their objective and showed the resulting AI income taxes can improve equality over classic tax models. Standard economic works often use the utilitarian objective (sum of all agent rewards). An alternative is the Rawlsian objective (social welfare is the reward of the lowest-income agent). We emphasize the choice of social welfare is flexible and a choice made by the designer(s) and users of the framework.

Another key example is the discount factor used to weight rewards over time. Whether to emphasize short-term vs long-term rewards is a social choice that has ethical implications. For example, firms may emphasize short-term profits over long-term health issues, which may disparately impact different social groups.

**Choice of agent model.** The behavior of agents is determined by the policy model, e.g., the neural networks used in our work. Neural networks are universal function approximators, given enough width (or depth) in their layers. However, in practice, neural networks may still encode structural biases and only parameterize a particular subspace of all theoretically possible policy models. For our networks, a particular concern might be architectural constraints: our policy networks are not recurrent (so only consider the current state) and sometimes don't allow correlated actions. With enough parameters these networks are still capable of representing a wide range of policies, but these architectural constraints represent implicit priors which conceivably might not reflect human decision-making. As such, more research is needed on what the limits are of neural networks in terms of emulating human behaviors, and to what extent more and diverse datasets can help alleviate such concerns.

**Choice of algorithms and learning strategies.** The RBC model is an economic "game" that has multiple equilibria. It is not well understood theoretically to which equilibria a given RL algorithm converges. Indeed, previous work has studied MARL beyond independent learners, including Nash-Q (Hu & Wellman, 2003), WoLF (Bowling & Veloso, 2002), and MADDPG (Lowe et al., 2017). This extends to our use of structured curricula, reward shaping, and other forms of multi-agent learning algorithms or strategies. These methodological choices can all impact the equilibria one finds (or doesn't find) using ML.

This is important because different equilibria can have different levels of social welfare and granular social outcomes (e.g., equality, type of work performed, unemployment level). From an ethical point of view, it is therefore possible that certain choices of algorithms, etc, may bias policy recommendations and simulation outcomes to socially or ethically undesirable situations. For instance, certain social groups in the simulation may be disparately impacted by policy recommendations. Therefore,

it is important for future research to analyze how different RL algorithms may selectively converge to certain equilibria, and how one might enumerate all possible equilibria. This is still a significant theoretical and empirical challenge.

Defining and justifying the objectives for the social planner and other methodological choices is a complex discussion, and requires a more in-depth understanding of the functioning of ML that is beyond the scope of this work. This requires multilateral, interdisciplinary discussion on, for example, what the preferred social choice is with respect to the definition of social welfare and constraints.

**Choice of hyperparameters.** RL algorithms may converge to different solutions depending on the chosen hyperparameters, e.g., learning rate, entropy regularization, or discount factor. For instance, the level of entropy regularization regulates the exploration-exploitation trade-off in actor-critic methods, a form of on-policy RL as used in our work. It is known that actor-critic methods may get stuck in suboptimal local maxima. This issue may be exacerbated in the multi-agent setting, where there are multiple equilibria, and it is unknown how algorithms converge towards different equilibria. As such, it is possible that certain choices of hyperparameter can encode structural biases towards certain outcomes in the simulation. These potential limitations are an area for future research.

**Robustness of Deep RL.** A key question is how robust learned policies are to perturbations in the simulation (parameters). This has ethical implications: policies that do well in simulation, may not do well in the real world if, e.g., income distributions differ between sim and real. As such, simulations that are not representative (enough) may lead to policy recommendations that disadvantage underrepresented social groups in the real world. More generally, it is well-known that deep learning models and RL policies can be very brittle and may not generalize well to unseen environments. As such, more robustness analysis should be done on any policy recommendation that is based on deep RL and related methods.

**Explainability and Simplicity of AI policies.** Even though AI policies may be effective, they may use intricate, unexplainable patterns in their input data to achieve high performance. Moreover, their behavior may vary wildly between different inputs. As a hypothetical example, an RL agent may make significantly different tax rate recommendations for people with slightly different income or education levels. Such behaviors can disproportionally affect underprivileged social groups, and have unintended short/long-term economic consequences, especially if models are not well-calibrated. It is an open question on what level/amount of data, or specific policy constraints, could mitigate such potential risk and harm. We also note that if one wanted to restrict the class of policies to ones that are sufficiently explainable, the same model-free policy optimization scheme could still be applied. Indeed, this is a big potential advantage of the RL approach.

## D MORE DETAILS ON THE RBC

**Settings.** Table 2 lists all the simulation parameters. Table 3 lists all the training hyperparameters.

| Parameter | Symbol | Values |
|---|---|---|
| Labor disutility | $\theta$ | 0.01 |
| Pareto quantile function scale parameter | - | 4.0 |
| Initial firm endowment | $B$ | 2200000 |
| Export market minimum price | - | 500 |
| Export market maximum quantity | - | 100 |
| Production function values | $\alpha$ | 0.2 to 0.8, increments of 0.2 |
| Initial capital | $K$ | 5000 or 10000 |
| First round wages | $w$ | 0 |
| First round prices | $p$ | 1000 |
| Initial inventory | $y$ | 0 |

Table 2: **Simulation Parameters.**

More implementation details follow:

| Parameter | Values |
|---|---|
| Learning Rate | 0.001 |
| Learning Rate (Government) | 0.0005 |
| Optimizer | Adam |
| Initial entropy | 0.5 |
| Minimum entropy annealing coefficient | 0.1 |
| Entropy annealing decay rate | 10000 |
| Batch Size | 128 |
| Max gradient norm | 2.0 |
| PPO clipping parameter | 0.1 or 0.2 |
| PPO updates | 2 or 4 |
| Consumer reward scaling factor | 5 |
| Firm reward scaling factor | 30000 |
| Government reward scaling factor | 1000 |

Table 3: **Training Hyperparameters.**

- For consumers, consumption choices range from 0 to 10 units for each good and work choices from 0 to 1040 hours in increments of 260.

- Consumers have a CRRA utility function with parameter 0.1, and a disutility of work of 0.01.

- For firms, price choices range from 0 to 2500 in units of 500; wage choices from 0 to 44 in units of 11.

- The 10 firms are split into two groups, receiving either 5000 or 10000 units of capital. Within these groups, firms receive a production exponent ranging from 0.2 to 0.8 in increments of 0.2. Thus each firm has a different production "technology".

- Firms invest 10% of their available budget (if positive) in each round to increase their capital stock.

- Government taxation choices range from 0 to 100% in units of 20%, for both income tax and corporate tax rates.

- The government can either value only consumers when calculating its welfare ("consumer-only") or value welfare of both consumers and firms ("total"), with firm welfare down-weighted by a factor of 0.0025 (to be commensurate with consumers).

- We set the minimum price at which firms are willing to export to be either 500 or 1000, and the quota for each firm's good to a variety of values: 10, 50, 100, or 1000.

- For consumers, consumption choices range from 0 to 10 units for each good and work choices from 0 to 1040 hours in increments of 260.

**Observation.** The environment *state* $s$ consists formally of all agent states and a general world state. Each agent observes can observe their own information and the global state:

$$s_{\text{global}} = \left(t, \{y_{i,t}\}_i, \{p_{i,t}\}_i, \{w_{i,t}\}_i, \{o_{i,t}\}_i\right). \tag{8}$$

Here $y_{i,t}$ is the available supply of good $i$, $p_{i,t}$ is the price, $w_{i,t}$ is the wage. The extra information $o_{i,t}$ includes whether good $i$ was overdemanded at the previous timestep and tax information.

In addition, consumer-workers observe private information about their own state: $(B_{i,t}, \theta)$ A firm $i$ also observes its private information: $(B_{i,t}, k_{i,t}, (0, \ldots, 1, \ldots, 0), \alpha)$, including a one-hot vector encoding its identity and its production function shape parameter $\alpha$. The government only sees the global state.

**Heterogeneity** We observe that although there is a fixed policy per agent type, agents exhibit heterogeneity through agent-specific input features, such as the disutility of work or production parameters.

## E    TRAINING DETAILS

For all experiments, we clip gradient $\ell_2$ norms to a maximum of 2.0, use a Huber loss for the value function head, and center and standardize estimated advantages. For PPO, we use the clipping based surrogate loss approach rather than a KL penalty, closely following the baseline implementation of Kostrikov (2018). We show pseudocode for a single simulation and training step in Algorithm 1 below.

---

**Algorithm 1** A single training step at time step $t$

---

$\pi_c, v_c \leftarrow$ consumer policy and value network
$\pi_f, v_f \leftarrow$ firm policy and value network, prices and wages masked according to $t$
$\pi_g, v_g \leftarrow$ masked government policy and value network, tax rates masked according to $t$
$\theta(t)$: disutility of work parameter, annealed over training steps
$w(t)$: entropy parameter, annealed over training steps according to $\max(\exp(\frac{-t}{\text{decay rate}}), 0.1)$

$s_c, a_c, r_c, s_f, a_f, r_f, s_g, a_g, r_g \leftarrow$ EnvironmentSimulate$(\pi_c, \pi_f, \pi_g, \theta(t))$
$\pi_c, v_c \leftarrow$ PPOUpdate$(\pi_c, v_c, s_c, a_c, r_c, w(t))$
**if** $t > t_{\text{start firm}}$ **then**
    $\pi_f, v_f \leftarrow$ PPOUpdate$(\pi_f, v_f, s_f, a_f, r_f, w(t - t_{\text{start firm}}))$
**end if**
**if** $t > t_{\text{start government}}$ **then**
    $\pi_g, v_g \leftarrow$ PPOUpdate$(\pi_g, v_g, s_g, a_g, r_g, w(t - t_{\text{start government}}))$
**end if**

---

## F    IMPLEMENTATION DETAILS

**Budget Constraints.**    We implement budget constraints on the consumers by proportionally scaling down the resulting consumption of all goods to fit within a consumer's budget. Thus, the consumer actions really represent attempted consumption – if the budget is small or stock is limited, the actual consumption enjoyed by the consumer may be lower. Firm budgets are allowed to go negative (borrowing money). However, because the firm's goal is to maximize profit, they are incentivized to take actions will will be profitable, increasing their budget.

**Scaling of Observables.**    The scales of rewards and state variables can vary widely in our simulation, even within time steps in a single episode. If the scales of loss functions or input features are very large or small, learning becomes difficult. We directly scale rewards and some state features by constant factors. For certain state features which have very large ranges (item stocks and budgets) we encode each digit of the input as a separate dimension of the state vector.

## G    GPU IMPLEMENTATION

We followed the WarpDrive framework (Lan et al., 2021) to simulate our DGE and do RL on a single GPU. We implemented the environment dynamics of our model as a CUDA kernel in order to increase the speed at which we can collect samples. We assigned one thread per agent (consumer, firm, or government); the threads communicate and share data using block-level shared memory. Multiple environment replicas run in multiple blocks, allowing us to reduce variance by training on large mini-batches of rollout data. We use PyCUDA (Klöckner et al., 2012) to manage CUDA memory and compile kernels. The policy network weights and rollout data (states, actions, and rewards) are stored in PyTorch tensors; the CUDA kernel reads and modifies these tensors using pointers to the GPU memory, thereby working with a single source of data and avoiding slow data copying.

