# OpenReview forum: "Finding General Equilibria in Many-Agent Economic Simulations using Deep Reinforcement Learning"
_ICLR.cc/2022/Conference — ICLR 2022 Submitted_

### Official Review · Reviewer_yaWE · 2021-10-25

**Correctness:** 3
**Technical Novelty And Significance:** 2
**Empirical Novelty And Significance:** 2
**Recommendation:** 5
**Confidence:** 3

**Details Of Ethics Concerns:**

The authors propose and study an economics model of a realtime business cycle. This model needs to be vetted to understand potential issues that might stem from economic-policy recommendations that it suggests.


**Main Review:**

**Review**

WIth recent advancements in Deep MARL, the door has opened to allow it to be brought out and applied to real world scenarios. In this submission, the authors look at economics based on an Real-Business-Cycles (RBC) model. I find this to be an exciting direction for research as it promises to help inform and improve economic policy design. That being said, the RBC model presented here is not motivated nor verified (why should it capture the important economic mechanisms of interest?). However, this may be because I am not adequately familiar with this literature (as I would expect the same of the average ICLR reader). The core contribution of this work is a reward shaping schedule to prevent discovery of "uninteresting" equilibria. Very little is said about the implications of the different shaping choices and there are no ablations or analysis of their component effects. As the conclusions, and thus take-aways for economic governance, are critically impacted by the bias imposed from this reward shaping I would have preferred to have it treated more rigoursly. Finally, I found the author's use of empirical game theoretic analysis to be essentially absent despite it being labelled as underpinning the results. In total, I am excited to see interest in this area; however, I do not think this particular unit of research is ready for publication at this time.

**Major Comments**

- Reward shaping is used in this work to bias equilibrium selection towards "non-trivial" solutions. Why are these non-trivial solutions not worthy of discussion if they are afforded by the mechanism? How is an equilibrium identified as trivial, and which qualities of solutions where the authors biasing against through their reward shaping?
- The ICLR community is not economics focused, and as a result it would be beneficial to include additional motivations and explanations when bringing in outside ideas. In particular, a large portion of this paper describes the implementation of a particular RBC model. It would be helpful to either punt a justification of its design to related work that uses the same mechanism. Or, if this is a novel mechanism, compare and contrast it with a pre-existing model include motivation and justification for deviations.
- As a cross-disciplinary work, I would like to see additional connections back to what is known about RBCs from econonmics. Are there known equilibria for this model? How do the results compare with economics solutions? This would allow an appreciation of the results for non-experts.
- Empirical game theoretic analysis is a method for solving games by constructing meta-games covering restricted strategy sets of the underlying-game, and solving the meta-game as an approximation of the underlying-game. In this work, convergence is measured by calculating the regret against the opponent's policy by freezing it and using Deep MARL to learn an approximate best-response. If all player's have no regret, then an e-NE is found. While this is all technically true, the quanlity of the analysis is weak, and as a result its likely that e is extremely large. Notably, the meta-game being analyzed in this setting is an incomplete 2x2 (abstracting to 2 players) NFG. The top-left cell being the payoffs being the pre-frozen joint-strategy performance, and the off-diagonal being the the approximate best-response performance against fixed policies. Could the authors provide evidence why such as imple empirical game is sufficent to ensure that e is small, and thus the meta-game's solution is at all relevant to the underlying game?
- The ethics statement does not seriously consider the ramifications of this work. Please discuss the biases and assumptions baked into this work and how this may lead to downstream issues instead of making blanket staements "should not be used to make deicsions in real-world systems" --- if this statement is true, then what's the point of this work?  It is important to think of the unknown-knowns that readers or future practitioners may overlook in the future.
- Sec 5, Par 1, "Empirical game-theoretic analysis .... with heterogeneous agents." This is not true. A meta-game is a choice of the practioner and through that choice they add restriction such as role-based meta-games often to gain performance benefits. EGTA is not restricted in analysis to any class of games, and can be analyzed through the lens of any game -- where certain choices, ala NFG, scale poorly (although this is a limitation of game theory that is simply inherited by empirical game theory).

**Minor Comments**

- Consumer reward has very large error bars in Fig 3, is there a particular explanation?
- The authors note that their method finds only low-welfare equilibria. What does the discovery of these equilibria in RBC suggest to policy-makers? Do the authors suspect this is an artifical resultant of reward shaping?



**Summary Of The Paper:**

This work investigates the use of Deep MARL in order to study economies. They propose and implement an RBC (economic market model) and propose a reward shaping schedule to bias agents into learning non-degenerate joint-strategies. In their setting, low-welfare equilria and provide a brief quantitative and qualitative analysis of discovered equilriba in open- and closed-economies.

**Summary Of The Review:**

I recommend rejecting this paper in a large part because (a) the main contributions (RBC model/reward shaping) have unclear/unspecified novelty, (b) weaknesses in the analysis described in the main review, and (c) need for contextualizing this work in both the MARL and economics literature (help the MARL community understand that this work is important!).

---

> ### Author Response · Authors · 2021-11-16
> **Response to reviewer (part 1)**
>
> Thanks for your comments. Please also see the general comments for context.
>
> **“Reward shaping is used in this work to bias equilibrium selection towards "non-trivial" solutions.”**
> We emphasize that we do not use reward shaping in this work. We use structured curricula that stagger training of the different agent types. We do gradually increase the penalty for disutility of work over time, but it reaches its final maximum value relatively early in the training schedule. We also use entropy regularization to encourage exploration, which is a standard technique in policy-based RL. Hence, towards the middle of training, the agents only see the semantically meaningful economic reward, and not other forms of reward that are meant to steer learning. As such, we do not deform the equilibria of the underlying economic model.
>
> **“Why are these trivial solutions not worthy of discussion?”**
> Trivial solutions involve the economy “shutting down”, where (almost) no work is done and thus no goods are produced or consumed. A large part of the purpose of the training schedules and other modeling choices was to avoid this type of trivial equilibrium and find other, non-trivial equilibria. For example, we start with random firm policies and gradually anneal the action masking, so that consumers can learn to correctly balance work and consumption over many price and wage levels before the firms start trying to optimize price and wage levels. We also gradually anneal the disutility of work, so that consumers won’t immediately be penalized for working. A simpler form of such training schedules were required for convergence to non-trivial solutions in previous work, e.g. Zheng et al. 2020.
>
> **“The ICLR community is not economics focused, and as a result it would be beneficial to include additional motivations and explanations when bringing in outside ideas.”**
>
> Please also see the general comments for a discussion on this point. RBC models are widely accepted dynamic economic models. RBC models are a large family of models involving representative worker-consumers, firms, and sometimes governments. Our model shares these characteristics but generalizes the typical RBC-style-model in several ways -- by allowing more heterogeneity among agents, by using discrete time steps and action spaces, and, crucially, by eliminating market-clearing constraints. A key advantage of our learning-based approach is that such (unrealistic) modeling constraints are not necessary anymore -- traditional models use these constraints for analytical tractability.
>
> These also represent differences from previous RL-for-economic-models work, which only learn policies for a single agent type, use continuous actions, and still impose many analytic constraints. Our approach also fits into the general framework of agent-based modeling; for a survey of agent-based economic models, see “Drawing on different disciplines: macroeconomic agent-based models” (Haldane and Turrell).
>
> **“... what is known about RBCs from econonmics. Are there known equilibria for this model? How do the results compare with economics solutions?”**
>
> It is known that RBC/DSGE-style models often admit multiple equilibria, see for one example "Some unpleasant properties of log-linearized solutions when the nominal rate is zero.” Braun et al. Typical solution methods for these models end up choosing one equilibrium arbitrarily in order to make a solution tractable. Our model is related to this type of model, but departs from them in several ways (no market clearing constraints, more heterogeneity of agents) that make solutions even less tractable. This is exactly where our work innovates: we show that multi-agent RL can learn, and can learn to find more than one non-trivial equilibrium.
>
> **“... what is the point?”:**
>
> Our work focuses on how to use RL and machine learning methodology to analyze complex economic models. This is of substantial interest and value, as RL gives a constructive approach to learn well-performing economic policies and analyze the behavior of the economy in response. Moreover, the use of learning means the economic model can be very flexible (e.g., don’t have to use unrealistic market-clearing constraints). However, it is still scarcely used in economic analysis and policy design, as the intersection of economics and ML is an emerging field.
>
> Methodologically, it is not straightforward to apply multi-agent RL to economic simulation. Therefore, our paper studies how to use structured curricula to “make it work” at all, as training often collapses when using a vanilla MARL approach. As such, using stylized simulations has great value to prove the merit of using RL.

---

> > ### Author Response · Authors · 2021-11-16
> > **Response to reviewer (part 2)**
> >
> > **Fig 3 error bars:** The plot shows the mean over 3 training runs, while the error bars represent the min and max. The training trajectories diverge somewhat around episode 100k, but by the end of training have ended up being near each other in reward. Note that in general, we found that solutions could converge to multiple qualitatively different equilibria.
> >
> > **“more ethics discussion”:**
> > We agree that future use of simulations for policy making requires careful inspection of the simulation, policies, and underlying data. We will extend the discussion, see below.
> >
> > Economic simulation enables studying a wide range of economic incentives and their consequences, including models of stakeholder capitalism. However, the version of the simulation as used in this work is not an actual tool that can be currently used for policy making.
> > While the current version of the economic simulation provides only a limited representation of the real world, we recognize that future, large-scale iterations can still contain biases and unrealistic assumptions. Furthermore, non-representative simulation environments may result in biased policy recommendations. For instance, the under-representation of communities and segments of the work-force in training data might lead to bias in simulations that build on those and lead to biased AI policies.
> >
> > Our RBC model is a stylized model of real economies. It contains assumptions and stylizations, and future simulation may miss (un)known features that pertain to equity and equality in the economy. We list a few salient features and assumptions below, although we cannot exhaustively enumerate all features that may be relevant in future research.
> >
> > Our RBC model features consumers that differ in skill and perform different amounts of work. However, we do not model more fine-grained distributional features, such as educational attainment, wealth, inheritance, geography, or others.
> > Similarly, firms produce a single good and can invest and pay wages. Any worker can work for any firm. We do not model hiring practices, the geographic location of firms, non-monetary incentives or benefits (e.g., health insurance). To accurately model inequity in the real world, including such features may be necessary.
> >
> > On the government and societal level, we model tax policies and simple redistribution of tax revenue. We do not model targeted redistribution, tax credits, application-specific subsidies (e.g., education support). We do not model trading, inflation, debt, and other macroeconomic features that may impact social groups disparately.
> >
> >
> > **“Finally, I found the author's use of empirical game theoretic analysis to be essentially absent despite it being labelled as underpinning the results.”**
> >
> > We’d like to clarify that **we do not use EGTA algorithms** (e.g., see “Evaluating Strategy Exploration in Empirical Game-Theoretic Analysis”, Yongzhao Wang, Qiurui Ma, Michael P. Wellman, 2021). Our main algorithmic contribution is showing that structured curricula stabilize learning and enable convergence to meaningful equilibria. By “empirical”, we refer to the fact that we use RL to learn well-performing policies, rather than deriving them analytically or proving theoretical results.
> >
> > As you point out, EGTA methods can in theory be generally applied to any game. However, it depends on whether an appropriate meta-game and meta-policies can be defined. Moreover, it is difficult to compute equilibria across any given set of policies. For instance, when two or more players can respond to a fixed agent, the best response consists of (a set of) Nash equilibria of a smaller game with the best responders. Finding those multi-agent best responses is computationally hard.
> >
> > To address this issue, we analyzed the best response of a single agent type (e.g., all consumers) to the fixed policies of the other agents (e.g., all firms and the government). This amounts to a Nash equilibrium analysis over agent types, where each agent type is a single player for the Nash equilibrium. We can perhaps think of the game over those players as a meta-game for the underlying game where each individual consumer/firm etc is a single player.
> >
> > However, we do not use EGTA (-inspired) methods, such as PSRO (Tuyls, 2018, https://dl.acm.org/doi/10.5555/3237383.3237402), to find equilibria using iteratively growing sets of policies, as these are hard to apply to our setting.
> > We will clarify this and correct the use of the label “empirical game theory”.

---

> > > ### Author Response · Authors · 2021-11-17
> > > **Response to reviewer (part 3)**
> > >
> > > **"While this is all technically true, the quanlity of the analysis is weak, and as a result its likely that e is extremely large.”**
> > >
> > > To check how large the e is for our empirically found equilibria we fixed the policies of all agents except one type of agent and let the RL find an approximate best response for this policy. We found that in most of the settings the improvement was **very small**, i.e., less than 1% of the reward obtained before (see Figure 4, middle). In 4 out of 10 instances, there is no significant improvement in the reward for all the agent types. These instances can be thought of as something like e-Nash equilibria with small e. In other instances, we observe some improvements in the best responding agent type, although these are still smaller than at the beginning of training. For example, in most of the instances, the improvement in the firm’s reward from best response was less than 5% of the improvement in its reward during training. In the case of consumers and governments, this improvement was less than 0.2% and 0.1% respectively. We will clarify this by adding Table 1 to the paper.
> > >
> > > **“Notably, the meta-game being analyzed ...  to ensure that e is small, and thus the meta-game's solution is at all relevant to the underlying game”**
> > >
> > > We ran single-agent RL for 100k steps to convergence to compute the best response for every agent type. This analysis showed that the unilateral best response only gives a small change in the objective, so e is empirical small (see Figure 4, middle). Of course, this is not a guarantee that we will always find a small e. However, there are no theoretical guarantees for this in the general case, and it is outside the scope of this paper to prove such guarantees.
> > >
> > > Although the e is non-trivial for these, the simultaneous RL appears to have stabilized at these policies and hints towards a notion of equilibrium other than Nash equilibrium. Using the 2x2 analogy given by the reviewer, if the stabilized policies are top left, and one of agents switches its action leading to an off-diagonal cell, it can cause the other agent to switch its action too, which could in turn lead to both agents switch their actions again leading back to the initial top-left cell. The simultaneous RL framework is rich enough for such dynamics to take place. Thus, the RL framework should be seen as a way to discover e-Nash equilibrium with small e which can be verified using the best response analysis. In other instances, further analysis can lead to interesting stability criteria other than Nash equilibrium as described above.

---

> > ### Comment · Reviewer_yaWE · 2021-11-17
> > **Reviewer Response**
> >
> > Overall, I have hesitations that this work still lacks (a) a reasonable bridge for the ICLR community to appreciate the economics of this work, and this is where the contributions of this work lay, and (b) absence of limitations and ethical discussion.
> >
> > I believe I see now that the authors do not want to claim the MARL style of training and equilibrium selection as a contribution of this work, but instead hope to focus it on the reward shaping and applications. This is fine, especially because I remain skeptical about the use of co-learning and deviation checking as a robust indicator of equilibrium discovery (I believe the reward shaping was absolutely critical for this to avoid the many uninteresting/degenerate solutions this method would discover). However, this puts more onus on appealing the narrative to the ICLR community (esp., ethics & limitations).
> >
> > **Reward shaping**
> > I respectfully disagree with your rebuttal. Reward shaping vs structured curricula is merely a point of pedantics. Curricula implies that the reward shaping is dynamic over the course of training, and is the point you raise here; however, nothing about reward shaping itself prohibits it from training. Importantly, the agent is trained with a reward signal that differs from the true "goal" reward signal. While you do indeed follow classical exploration methods mid-learning, you cannot claim and verify that the early training with the shaped reward has not constrained your final achievable policy space. It's not necessary that work meets this requirement, but it is worth acknowledging as a limitation. Any biases that might influence final policies is important in interpreting the results, especially in a domain that I imagine is long-term meant to influence the real world.
> >
> > **Trivial Solutions / Economics Focus / RBCs**
> > Thank you for the comments on the trivial solutions and RBC background. Discussions like this are useful within the work or supplementary material to help understand methodology. My ICLR community comment was intended to encourage adding this motivation within the paper itself. I think this is critically important to the success of this work at ICLR.
> >
> > **Ethics**
> > I understand that economics & ML is emerging, but in my opinion, this means that it is even more important to really focus on the future ethics of the area now. The addition of the ethics review as a field of the review process is meant for this purpose explicitly. The choice of the environment (esp. reward function), RL algorithm, policy class, etc. all play an important but difficult-to-understand role in equilibria selection. I do not think this work being a proof of concept exonerates this work a full ethical discussion about the choices made within. This does not mean that you're on the hook with considering every permutation of RBC model so that can consider everything "fairly"; nor, are you responsible for defending RBCs themselves. Punt RBC justifications to pre-existing work, discuss any changes you've made and what their implications are. Focus on economics (RBC) & ML and have a discussion about it.
> >
> > **EGTA / Convergence**
> > My apologies, I did not mean to imply that this work uses an EGTA/PSRO-style algorithm. I only meant to use their frameworks for an analogy.
> >
> > The size of e is not only measurable by using an approximate best response (ABR) to measure payoff from deviation. The "approximatiness" of the BR method also influences the size of e. This is particularly important when using RL, because it's particularly fragile to hyperparameter selection. The joint frozen-opponent-set and environment each constitute a unique black-box single-agent environment, potentially posing vastly different optimization problems. This is a problem with MARL in general, not this particular work. It does undermine the credibility of these claims about the quality and efficacy of the equilibria discovered. How are the hyperparameters selected? Is the curriculum used here? Does the curriculum make sense to also use here when the opponent is already shaped? This is a great place to tie into the ethics/limitation discussion that there are clear limitations to this approach; in particular, the fragility of DL and RL (especially combined).
> >
> > I agree that theoretical guarantees are out of the scope of this work, and have no expectation that this work should include any quantifiable bounds on e.

---

> > > ### Author Response · Authors · 2021-11-18
> > > **Modified ethics discussion**
> > >
> > > We’d like to point out that we already extended the ethics discussion in our previous response to your review. So far, it has focused on the choices and the limits of the economic modeling aspects of our work. We understand that we can add a more general discussion of the ethical implications of using ML for economics too. We would appreciate your feedback on the expanded discussion below. Please let us know if you have more suggestions on where to expand it, or where the wording should be adjusted.

---

> > > > ### Author Response · Authors · 2021-11-18
> > > > **Ethics discussion**
> > > >
> > > > **Assumptions, limitations, and ethical implications of using ML for economics.**
> > > >
> > > > All choices in the economic simulation model, RL algorithms, reward functions, etc, play an important but difficult-to-understand role in equilibria selection and policy design. As in all ML applications, there are assumptions and limitations in the methodology. This has ethical implications for their use in future policy design applications.
> > > >
> > > > **Mitigation strategies and interdisciplinary research.**
> > > >
> > > > Economic simulation enables studying a wide range of economic incentives and their consequences, including models of stakeholder capitalism. However, the version of the simulation as used in this work is not an actual tool that should be used for policy making.
> > > >
> > > > Many design choices influence the eventual policy recommendations. For example, the designer is free to set the social welfare objective that the government optimizes for. As such, it is crucial that these choices are debated and made in a socially acceptable fashion by all stakeholders, and made transparent and accessible to all.
> > > >
> > > > More generally, to mitigate ethical risk, further mitigation strategies may include performing a what-if analysis over worst-outcomes, opening research results to domain experts (social scientists, ethics experts, etc), and open-sourcing the research results, amongst others. In all, the design and use of ML for policy recommendations will require robust, multilateral discussion and careful consideration of ethical risk, potential harm, and which trade-offs are being made.
> > > >
> > > > We now detail some assumptions, limitations, and potential ethical risk among different dimensions of using ML for economics. We stress that there can be more (unknown) aspects that we do not address here. As such, we see this discussion as a starting point of discussion for the ML and economics community.
> > > >
> > > > **Economic simulation and data**
> > > >
> > > > While the current version of the economic simulation provides only a limited representation of the real world, we recognize that future, large-scale iterations can still contain biases and unrealistic assumptions. Furthermore, non-representative simulation environments may result in biased policy recommendations. For instance, the under-representation of communities and segments of the work-force in training data might lead to bias in simulations that build on those and lead to biased AI policies. As such, collecting more representative data is a key challenge for future research in using ML for economic policy recommendations.
> > > >
> > > > Our RBC model is a stylized model of real economies. RBC models are a commonly used class of economic models (see e.g. Smets & Wouters, “Shocks and Frictions in US Business Cycles: A Bayesian DSGE Approach” for one representative paper). However, as any model, it contains assumptions and stylizations. Future simulations may miss (un)known features that pertain, e.g., to equity and equality in the economy. Therefore, using simulations that are not representative or well-calibrated, can exacerbate or create new socio-economic issues.
> > > >
> > > > We list a few salient features and assumptions below, although we cannot exhaustively enumerate all features that may be relevant in future research.
> > > >
> > > > Our RBC model features consumers that differ in skill and perform different amounts of work. However, we do not model more fine-grained distributional features, such as educational attainment, wealth, inheritance, geography, or others.
> > > >
> > > > Similarly, firms produce a single good only and can invest and pay wages. Any worker can work for any firm. We do not model hiring practices, the geographic location of firms, non-monetary incentives or benefits (e.g., health insurance). To accurately model inequity in the real world, including such features may be necessary.
> > > >
> > > > On the government and societal level, we model tax policies and simple redistribution of tax revenue. We do not model targeted redistribution, tax credits, application-specific subsidies (e.g., education support). We do not model trading, inflation, debt, and other macroeconomic features that may impact social groups disparately.
> > > >
> > > > Our RBC is more general than commonly used models: we do not enforce market clearing, for instance. Market clearing is an unrealistic assumption that supply always meets demand. Economic theory uses such constraints to make analysis tractable. In contrast, our learning approach is flexible and does not require such simplifying assumptions.
> > > >
> > > > We also assume that all agents can observe the wages offered by all the firms. However, in the real world not all agents have equal access to information -- and this is a feature that can be studied by future research. As such, we view the flexibility of our learning approach as a positive, in that our framework may allow for studying more representative models.

---

> > > > > ### Author Response · Authors · 2021-11-18
> > > > > **Ethics discussion**
> > > > >
> > > > > **Choice of economic incentives and rewards.**
> > > > >
> > > > > Agents optimize their behavior given economic incentives, as modeled by their reward function. As such, future economic AI policies should clearly describe for which reward function they were optimized. Furthermore, more research is needed to understand how the choice of reward function influences the resulting policies, and how social and ethical values can be transparently encoded in reward functions. It is also an open question which ethical/social norms and values can or cannot be quantified, and how to encode trade-offs between conflicting values.
> > > > >
> > > > > For example, the planner optimizes its policy to maximize “social welfare”, a standard economics concept. However, the definition of social welfare heavily influences the resulting policy and social outcomes. For example, Zheng et al, 2020 used equality x productivity as their objective and showed the resulting AI income taxes can improve equality over classic tax models. Standard economic works often use the utilitarian objective (sum of all agent rewards). An alternative is the Rawlsian objective (social welfare is the reward of the lowest-income agent). We emphasize the choice of social welfare is flexible and a choice made by the designer(s) and users of the framework.
> > > > > Another key example is the discount factor used to weight rewards over time. Whether to emphasize short-term vs long-term rewards is a social choice that has ethical implications. For example, firms may emphasize short-term profits over long-term health issues, which may disparately impact different social groups.
> > > > >
> > > > > **Choice of agent model.**
> > > > >
> > > > > The behavior of agents is determined by the policy model, e.g., the neural networks used in our work. Neural networks are universal function approximators, given enough width (or depth) in their layers. However, in practice, neural networks may still encode structural biases and only parameterize a particular subspace of all theoretically possible policy models. For our networks, a particular concern might be architectural constraints: our policy networks are not recurrent (so only consider the current state) and sometimes don’t allow correlated actions. With enough parameters these networks are still capable of representing a wide range of policies, but these architectural constraints represent implicit priors which conceivably might not reflect human decision-making. As such, more research is needed on what the limits are of neural networks in terms of emulating human behaviors, and to what extent more and diverse datasets can help alleviate such concerns.
> > > > >
> > > > > **Choice of algorithms and learning strategies.**
> > > > >
> > > > > The RBC model is an economic “game” that has multiple equilibria. It is not well understood theoretically to which equilibria a given RL algorithm converges. Indeed, previous work has studied MARL beyond independent learners, including Nash-Q (Hu & Wellman 2003), WoLF (“Multiagent Learning Using a Variable Learning Rate”, Bowling et al.), and MADDPG (“Multi-Agent Actor-Critic for Mixed Cooperative-Competitive Environments”, Lowe et al.). This extends to our use of structured curricula, reward shaping, and other forms of multi-agent learning algorithms or strategies. These methodological choices can all impact the equilibria one finds (or doesn’t find) using ML.
> > > > >
> > > > > This is important because different equilibria can have different levels of social welfare and granular social outcomes (e.g., equality, type of work performed, unemployment level). From an ethical point of view, it is therefore possible that certain choices of algorithms, etc, may bias policy recommendations and simulation outcomes to socially or ethically undesirable situations. For instance, certain social groups in the simulation may be disparately impacted by policy recommendations. Therefore, it is important for future research to analyze how different RL algorithms may selectively converge to certain equilibria, and how one might enumerate all possible equilibria. This is still a significant theoretical and empirical challenge.
> > > > >
> > > > > Defining and justifying the objectives for the social planner and other methodological choices is a complex discussion, and requires a more in-depth understanding of the functioning of ML that is beyond the scope of this work. This requires multilateral, interdisciplinary discussion on, for example, what the preferred social choice is with respect to the definition of social welfare and constraints.

---

> > > > > > ### Author Response · Authors · 2021-11-18
> > > > > > **Ethics discussion**
> > > > > >
> > > > > > **Choice of hyperparameters.**
> > > > > >
> > > > > > RL algorithms may converge to different solutions depending on the chosen hyperparameters, e.g., learning rate, entropy regularization, or discount factor. For instance, the level of entropy regularization regulates the exploration-exploitation trade-off in actor-critic methods, a form of on-policy RL as used in our work. It is known that actor-critic methods may get stuck in suboptimal local maxima. This issue may be exacerbated in the multi-agent setting, where there are multiple equilibria, and it is unknown how algorithms converge towards different equilibria. As such, it is possible that certain choices of hyperparameter can encode structural biases towards certain outcomes in the simulation. These potential limitations are an area for future research.
> > > > > >
> > > > > > **Robustness of deep RL**
> > > > > >
> > > > > > A key question is how robust learned policies are to perturbations in the simulation (parameters). This has ethical implications: policies that do well in simulation, may not do well in the real world if, e.g., income distributions differ between sim and real. As such, simulations that are not representative (enough) may lead to policy recommendations that disadvantage underrepresented social groups in the real world.
> > > > > > More generally, it is well-known that deep learning models and RL policies can be very brittle and may not generalize well to unseen environments. As such, more robustness analysis should be done on any policy recommendation that is based on deep RL and related methods.
> > > > > >
> > > > > > **Explainability and Simplicity of AI policies.**
> > > > > >
> > > > > > Even though AI policies may be effective, they may use intricate, unexplainable patterns in their input data to achieve high performance. Moreover, their behavior may vary wildly between different inputs. As a hypothetical example, an RL agent may make significantly different tax rate recommendations for people with slightly different income or education levels. Such behaviors can disproportionally affect underprivileged social groups, and have unintended short/long-term economic consequences, especially if models are not well-calibrated. It is an open question on what level/amount of data, or specific policy constraints, could mitigate such potential risk and harm. We also note that if one wanted to restrict the class of policies to ones that are sufficiently explainable, the same model-free policy optimization scheme could still be applied. Indeed, this is a big potential advantage of the RL approach.

---

> > > ### Author Response · Authors · 2021-11-18
> > > **Further comments**
> > >
> > > Thanks for your reply. We’d like to work with you on improving and extending the narrative to address your concerns.
> > >
> > > a) We have extended the discussion of the significance and context of our work in economics, geared towards ML researchers. We have emphasized several aspects: RBC models are commonly used in economics and capture some fundamental aspects of real economies. We show that RL can explicitly find approximate (game-theoretic) equilibria in RBC-style models; traditional economics methods largely cannot do so in RBC models with the complexity that we studied. (We will add more discussion of where our model deviates from simpler RBC-type models, and why this makes RL valuable, in the section where the dynamics are defined.) To our knowledge, this is the first work that explicitly constructs approximate equilibria for these types of models, a first for economists. Moreover, our best response analysis suggests the found solutions are quite close to real equilibria, the reward improves by < 5% in most cases, for any agent type. And due to the relative generality of model-free RL, we would expect that this general approach would work for a wide variety of multi-agent economic models that have analytically intractable properties.
> > >
> > > b) we’d like to point out that we gave an extended ethics discussion in our previous response to your comment. The ethics discussion right now has focused on the economic modeling aspects, i.e., what aspects of the economy we do or don’t include. Per your suggestion, we have added more discussion about ethical implications of the ML methodology aspects. We hope this 2nd extension of the ethics discussion meets your expectations. It would also be helpful to understand what level of detail and further scope of discussion is sufficient to accept our paper, in your opinion. Also note that due to space constraints, the full ethics discussion is in the Appendix of the paper.
> > >
> > > **Reward shaping**
> > >
> > > We acknowledge that curricula/reward shaping, including entropy regularization, can influence which equilibrium we converge to (indeed, this is an explicit goal, to avoid the trivial equilibria) and may also limit the types of non-trivial equilibria we find. We have expanded the discussion of this in the ethics discussion -- also see below.
> > >
> > > We’d like to clarify that our learning curricula have two key features: 1) entropy regularization, 2) slowly making the full range of actions available, and 3) starting agent training at different times, intentionally delaying training the firms (second phase) and the government (third phase), to encourage stable learning. Specifically, our curricula do more than just “shaping rewards” (entropy reg).
> > >
> > > We also note that the agents are receiving the true “goal” signal for quite a while before the end of training. Using entropy regularization is often found necessary in actor-critic style RL. The entire policy space is available to the agents by the middle of training. However, we do acknowledge that the agent’s policy could be biased towards certain behaviors because of the curriculum.
> > >
> > > **Best Response Hyperparameters**
> > >
> > > The best response (BR) analysis amounts to training an agent type, e.g., all consumers, while keeping the other agents fixed. As such, it is a hybrid “multi-agent” RL problem where all the agents share the same model weights. We did not use scheduled training for the approximate best-response training, although we did use a constant level of entropy regularization. We did not tune approximate-BR-specific hyperparameters, but rather used the same training hyperparameters (learning rate, etc.) that were used to train the policies in the main multi-agent loop.  Despite using entropy regularization, we found that best responses policies tended to converge quickly, across many different scenarios.
> > >
> > > We found empirically that these results are quite robust. However, we do not have a theoretical guarantee, e.g., to upper bound the best response reward improvement. As pointed out, we believe this is out of the scope of this work.
> > > We do want to emphasize that this is a general concern with all MARL work, as you also mention, and in lieu of theoretical guarantees, a good proxy for “equilibrium” is the objective itself (e.g., social welfare) and inspecting the learned behaviors by the consumer, firms, and government -- for example, we observe firms choosing strategies that reflect the efficiency of their production functions. We have added a discussion of this to the ethics section.
> > >
> > > In a separate comment thread, we include a draft of a further expanded discussion of ethical impacts.

---

> > > > ### Comment · Reviewer_yaWE · 2021-11-23
> > > > **Reviewer Reply**
> > > >
> > > > I've been mulling it over, and I suppose my stopping point is that I am struggling to see the claim regarding convergence to an e-NE as necessarily interesting.
> > > >
> > > > Every joint-strategy is an e-NE, for some e. While this admits e to be so large that the statement is a bit vacuous, it also helps contextualize these claims. If nothing can be said about the bounds or consistency of e, then the claims of finding an e-NE can be similarly vacuous.
> > > >
> > > > This work uses DeepRL as an approximate BR method in order to estimate e. Even as an entrenched member of this field, I am highly skeptical of how well DeepRL can serve as an oracle, and such give a reasonable estimate of e. This is due to how stochastic and fragile the implementation and settings of DeepRL can be.
> > > >
> > > > Combining these two ideas, we're seeing that a method can converge to anything, and be seen as an e-NE when the oracle method is sufficiently bad. Therefore, it appears that the most influential piece of this method's ability to find an e-NE is the reward shaping. But then it's almost begging the question to claim/analyze particular discovered e-NE when what appears to be the key contribution drives the convergence to those points.
> > > >
> > > > I've increased my score, because I don't believe the onus should be on you to solve all of these complex problems, but I encourage the authors to consider these thoughts. Moreover, I appreciate the inclusion of a rigorous ethical discussion that was provided in the other thread.

---

### Official Review · Reviewer_pWNT · 2021-11-02

**Correctness:** 2
**Technical Novelty And Significance:** 2
**Empirical Novelty And Significance:** 3
**Recommendation:** 3
**Confidence:** 4

**Main Review:**

The most interesting point in this paper is that the proposed scheme breaks the curse of many agents. In the numerical example, the total number of agents is more than 100. As in economics models, a single agent usually represents a large population of homogeneous agents, 100 is a sufficiently large number from an application point of view. The success of the deep RL algorithm studied in this paper highly relies on WarpDrive, a recently developed framework that supports running both simulation and RL on a GPU.

Yet, as far as I'm concerned, running the algorithm successfully is just a first step for studying this fundamental problem. Although the authors reported their empirical results, they failed to provide enough theoretical guarantees for supporting the results.

(1) Can the proposed algorithm converge to the ($\epsilon$-)Markov equilibrium almost surely? If so, can you provide sufficient conditions for the convergence?

In this paper, the convergence to the equilibrium is checked empirically (p6-best responses and local equilibrium analysis). Yet, a specific case could not provide the convergence guarantee for the general setting. This specific case is relatively special: the 100 consumer-works and the 10 firms are homogeneous, respectively. Therefore, it is similar to (but not the same as) a potential Markov game, which is not significantly different from a single-agent MDP. In the study of dynamic general equilibrium, the heterogeneity of agents has received more and more attention. The convergence of the algorithm in the heterogeneous setting (e.g., the 100 consumer-works are different from one to another) could be more difficult.

(2) Can the proposed algorithm learns a real Stackelberg strategy?

In the proposed model, the government has a first-mover advantage as in a Stackelberg game. In the optimization literature, it is well known that finding the Stackelberg equilibrium is an NP-hard problem. When both the leader and the followers become MDP agents, this problem would definitely become more complicated. Yet, in this paper, no theoretical or empirical results could support that a (locally) Markov Stackelberg equilibrium is found. The only numerical result for this issue is provided in p7-comparing with baseline government policies: it is reported that the RL government policy improves the social welfare by 15% compared with the case with fixed tax rates. However, it doesn't mean that it is really a Stackelberg strategy unless sufficient results could support that the government has fully realized its first-mover advantage.

Meanwhile, the algorithm itself is not described clearly in this paper. For example, the trick for running both simulation and RL on a GPU is from WarpDrive, yet the details of the WarpDrive framework are not even introduced in this paper. The structured learning curricula and the action space annealing procedure, as another example, are introduced using textual description completely without providing any equations and technical details.

**Summary Of The Paper:**

This paper proposed a deep reinforcement learning framework for finding dynamic general equilibrium, which is one of the most fundamental problems in economics. As the dynamic general equilibrium is a special case of the Markov game, this problem has also been recognized as a significant topic in machine learning. The proposed scheme is tested in a real-business-cycle model with 100 worker-consumers, 10 firms, and a government, the scale of which is much larger compared with the numerical examples in most related works.

**Summary Of The Review:**

The paper studied a fundamental problem and tested an efficient algorithm for solving it. However, I do not think the paper can be accepted in the present form because (1) the success of the algorithm highly relies on a trick proposed in another paper; (2) besides running the algorithm empirically, no theoretical analysis of the proposed scheme is provided in the paper.

---

> ### Author Response · Authors · 2021-11-15
> **Response to reviewer**
>
> Our work is to our knowledge the first empirical application of multi-agent RL to economic simulation where all agent types learn. As stated in the general comments, our work shows how to make multi-agent RL learn non-trivial equilibria through structured curricula. Vanilla independent MARL typically collapses in these non-trivial economic simulations. Although we do not show theoretical guarantees (see below), the empirical demonstration that it works is non-trivial and to our knowledge, the first of its kind in the context of ML for economics. As such, we believe the contributions of this work stand on their own and proving theoretical guarantees is beyond the scope of this work.
>
> **GPUs are useful, but are not required:** The WarpDrive approach, of running not just the policy networks but also the environment simulation on the GPU, is useful, but it is not an absolute requirement for these models. One could implement all these techniques in the usual CPU-environment/GPU-policy way, it would just be slower and more computationally costly. We will add some clarifying explanations of how it works.
>
> **Theoretical guarantees:** It is common in deep RL to only provide empirical analysis, as theoretical guarantees can be difficult to come by. In general, truly testing for a global equilibrium would require globally optimizing over the space of all neural network policies. Instead, one can measure a numerical objective, e.g., social welfare, to measure the performance of the learned policy, and use empirical best response analysis to measure how close to a local equilibrium one is. As an analogy, the performance of an AI agent in >2-agent poker can be measured using its stack size (Superhuman AI for multiplayer poker, Brown, 2019).
>
> We don’t expect that our approach, as specified, would come with any formal guarantees of converging to equilibrium. Our setting is beyond the reach of current theory for a couple reasons: first, our use of deep neural networks; second, we use a >2 agent general sum game.
>
> The theory for when model-free policy optimization for MARL converges to Nash equilibrium is somewhat limited. Theorems generally either only apply to tabular policies, e.g., see “Actor-critic policy optimization in partially observable multi-agent environments”, Srinivasan et al., 2018, (in fact these particular theorems don’t come with a sublinear regret bound so can’t actually guarantee convergence to Nash on average). Other cases also apply only to very limited families of games, e.g., see “Policy Optimization Provably Converges to Nash Equilibria in Zero-Sum Linear Quadratic Games”, Zhang et al., 2019.
>
> **Our “game” is Stackelberg-like, but not entirely Stackelberg:** We also note that, while our model does have Stackelberg-like qualities (the government has a first-mover advantage, etc.), so that we describe it as “akin” to a Stackelberg game, our model is not a Markov Stackelberg game. As we understand it, this would refer to a Markov process where at each time step, the leader and follower observe a state, the leader chooses an action, the follower observes it and chooses their action, and then given both actions, the environment chooses the next state according to a transition kernel. In our model, by contrast, everyone chooses actions jointly and simultaneously! Moreover, consumer reward as a result of their action depends mainly on prices/wages which are observed in the current state, while firms and governments act to alter the next state. Additionally, while there is some shared global state, each agent type cannot observe the complete state available to the other agent types. As such, it is not correct to say that we need to find a Stackelberg equilibrium.
>
> **Homogeneous agents:** We note that the agents have different parameters associated with them (disutility of work, production parameters for the firms, etc.), so in that sense, they are heterogeneous.
>
> However, each type of agent has a single policy network. We see this as a kind of weight sharing, a common approach in multi-agent learning, e.g., as used in the DotA 2 teams of AI agents [1]. Agents that share weights can still model heterogeneity between agents, because the inputs to the policy can differ between agents. For example, one can condition policy models (that share weights) on variables that parameterize this heterogeneity, e.g., when workers have different disutilities of work, this parameter can be an input feature to the policy network when choosing each worker’s different action.
>
> Similar approaches are taken in other RL for economic modelling work, such as Zheng et al. 2020, Hill et al. 2021. We note that we do observe qualitatively different strategies depending on the agent’s parameters, see e.g. the prices and wages set by the different firms with different production parameters in figure 6.
>
> [1] Dota 2 with Large Scale Deep Reinforcement Learning, OpenAI, https://cdn.openai.com/dota-2.pdf.

---

### Official Review · Reviewer_AWjh · 2021-11-03

**Correctness:** 3
**Technical Novelty And Significance:** 3
**Empirical Novelty And Significance:** 3
**Recommendation:** 6
**Confidence:** 3

**Main Review:**

As pointed out by the authors, there are relatively few works on reinforcement learning for economics. This direction seems promising. However, I have a few questions:

1. In the abstract and the introduction, could you please clarify what you mean by finding optimal agent policies “explicitly” versus “implicitly” (top of page 2)?

2. In section 3, several ideas are described but there is no clear description of the algorithm. Since the problem considered here is rather complex, I recommend adding a description (e.g., pseudo-code) of the MARL method. It would be helpful for the reader to see how the ingredients described in this section are combined.

3. In section 4, at the bottom of page 6, how do you check the local equilibrium property for the leader? The text seems to indicate that you fix all the followers and train the leader alone. However, I thought that the leader's reward depends the followers' reactions. Are the followers' policies functions of the leader's policy? Here again, I recommend adding a clear description of the quantities that are computed for the plots.

4. Figure 4, left plot: Even at the end, there is still a significant improvement for the firms (roughly equal to the improvement at the middle of the training). How can you explain that?

**Summary Of The Paper:**

This paper proposes a multi-agent deep reinforcement learning (DRL) method to compute general equilibria in economics. The main contribution is hence algorithmic. The method uses a combination of DRL, structured learning curricula, and suitable annealing of action space and of some penalty coefficients (which are useful to make some problems easier to solve, but distort the computed solution). The authors apply their method to an example of real-business-cycle model, which involve three types of agents: firms, consumers and a government. The problem is of Stackelberg (or leader-follower) type, with the government acting as a leader. Computing equilibria for such problems is generally very challenging.


**Summary Of The Review:**

The application of multi-agent reinforcement learning for economic problems seems very promising, particularly given the high complexity of these problems. The main contribution is algorithmic, but without a precise description of the algorithm, I find it hard to really understand the method and to assess the impact this work could have.

---

> ### Author Response · Authors · 2021-11-16
> **Response to review**
>
> **Question 1 : “Explicit vs Implicit”.**
>
> We find policies explicitly using RL. By “implicit”, we mean, for example, that the optimal policy can be characterized by the Bellman equation. However, solving the Bellman equation is very challenging generally. RL is a computational solution to try to solve it (approximately)), where analytical methods often fail to find the solution in complex settings, e.g., with high-dimensional state space of nonlinear environment dynamics.
>
> These barriers can be seen in the economics literature. Analytical work in economics has studied implicit characterizations of the impact of taxes, for instance. One can derive analytical expressions for the distortion in consumer savings or capital due to a taxation policy [1] or the asymptotic behavior of taxes in the far future, the tax rate on the highest-skill agent, or the structure of incentives [2]. However, these methods generally *cannot* find the full tax policy explicitly or only describe part of it.
>
> [1] M. Golosov, M. Troshkin, A. Tsyvinski, Optimal dynamic taxes, National Bureau of Economic Research (2011)
>
> [2] D. Acemoglu, M. Golosov, A. Tsyvinski, Dynamic Mirrlees Taxation under Political Economy Constraints, The Review of Economic Studies, 77, 841 (2010)
>
> **Question 2: “pseudocode"**
>
> We will add pseudo-code in the appendix. Here is a draft of pseudocode for a training step at time t:
>
> ```
> \pi_c, v_c = consumer policy and value network
> \pi_f, v_f = firm policy and value network, prices and wages masked according to t
> \pi_g, v_g = masked government policy and value network, tax rates masked according to t
> \theta(t): disutility of work parameter, annealed up to true theta over training steps
> w(t): entropy regularization strength, scheduled to decrease over training steps as w(t)=\max(\exp(-t/decay rate), 0.1).
> s_c, a_c, r_c, s_f, a_f, r_f, s_g, a_g, r_g = EnvironmentSimulate(\pi_c, \pi_f, \pi_g, \theta(t), initial conditions)
> \pi_c, v_c = PPOUpdate(\pi_c, v_c, s_c, a_c, r_c, w(t))
> If t > t_{start firm}:
> \pi_f, v_f = PPOUpdate(\pi_f, v_f, s_f, a_f, r_f, w(t - t_{start firm}))
> If t > t_{start government}:
>     \pi_g, v_g = PPOUpdate(\pi_g, v_g, s_g, a_g, r_g, w(t - t_{start government})
>  ```
>
> **Question 3:**
> The followers’ actions within a simulation episode depend on the state, which is impacted by the leader’s actions in previous time steps, and follower actions do impact leader reward. However, the follower’s policy network parameters are not a function of the leader’s policy network parameters -- as far as the training dynamics go, all are trained jointly. As such, when doing the approximate best-response analysis, we train policies independently.
>
> We will clarify the plots. At the bottom of section 4, the units of the social welfare score that is improved by the government refer to the total utility over all consumers, plus the total profit of all firms scaled by 0.0025 (to be commensurate with consumers).
>
> **Question 4:**
> It’s an approximate equilibrium. While the firm can improve its objective at the end of training, this improvement is still much smaller than it was at the beginning of training. For example, the firm’s reward improves from almost zero (or even negative values) to more than 10^5 during training. On further training the firms while keeping the other agents’ policies fixed, we get an improvement that is less than 5000. Thus the improvement achieved by any additional training is less than 5% of the improvement achieved during training. Similarly, the improvement achieved by additional training for consumers and governments is less than 0.2% and 0.1% respectively. (See Table 1 in the general comment.) Although the improvement for firms is larger than that for consumers and governments, an improvement of 5% is still relatively small. We will clarify the scale of change in the paper.

---

> > ### Comment · Reviewer_AWjh · 2021-11-28
> > **Reviewer reply**
> >
> > I would like to thank the authors for their response. I have increased my score, although I still believe that the presentation of the paper should be improved before publication if accepted.

---

### Official Review · Reviewer_JN4A · 2021-11-03

**Correctness:** 4
**Technical Novelty And Significance:** 3
**Empirical Novelty And Significance:** 3
**Recommendation:** 5
**Confidence:** 3

**Main Review:**

Using a multi-agent reinforcement learning algorithm to solve the complicated game considered in the paper is natural and reasonable. The structured learning curricula runs in multiple stages and could potentially ease the problem of instability in the training process. Also, it is interesting to see that the GPU-only framework is able to significantly accelarate the training process. The experiment results look convincing and interesting.

There are a number of notations that are used before they are defined (for example, p_i, J, \tau), making the paper hard to follow. I have to read back and forth, and keep guessing the meaning of these symbols. The readability can be significantly improved.

**Summary Of The Paper:**

This paper uses a multi-agent reinforcement learning algorithm to simulate an economic environment and solve the general equilibrium of the induced game. The authors propose to use a structured learning curriculum that runs only on GPUs. The authors conduct experiments to show that their algorithm converges fast and that the solution represents an epsilon Nash equilibrium.

**Summary Of The Review:**

In general, I find the paper interesting. But the readability can still be improved.

---

> ### Author Response · Authors · 2021-11-15
> **Response to review**
>
> We want to clarify that our methods do not absolutely require GPUs. However, as we have many RL agents (~100), we found empirically that using a GPU significantly speeds up simulation and getting empirical results, e.g., over 10x faster than with using CPUs.
>
> We can clarify the flow of definitions and variables to improve readability. Currently a lot of the definitions and exposition are in the appendix due to space constraints. We’ve made sure that all variables, especially in the section describing the model dynamics, are declared in the text before being used. Please check back in a few days for an updated manuscript.

---

### Author Response · Authors · 2021-11-15
**General comments**

Thank you all for your thoughtful reviews. Most reviewers seemed satisfied with the empirical experiments and results. As a key concern, all reviewers raised concerns about the exposition. To address this, we will further clarify sections that are unclear, e.g., around the economics context of this work, and improve the clarity of the presentation and significance of the results. Some theoretical complaints were raised as well: while these are thought-provoking, we mostly see theory as outside the scope of our paper, which deals with a problem setting beyond the reach of current theory.

**Given the interdisciplinary nature of our work, we’d like to clarify our contributions:**

We use RL to find solutions to an unusually complex, heterogeneous, multi-agent economic model. A key innovation is that we can abandon many unrealistic assumptions usually made by economists for analytical tractability, yet still learn economically plausible solutions. There is some previous work in this area, but it generally only treats a single agent type as a learner, or does not deal with complex economic relationships as in our model. The RBC model here is chosen to be related to standard economic models -- minus some tractability assumptions -- but the most techniques and methods we present here are general. Our results motivate using learning-based tools for dealing with similarly complicated and heterogeneous economic models in other areas.

On the ML side, RL, especially multi-agent RL, rarely “just works”. An important contribution of our work is showing that the use of structured curricula can ensure training does not collapse. Our environment implementation on the GPU, while not strictly required and based on techniques presented elsewhere, also greatly accelerates our ability to collect samples and train successfully. We see our contributions here as practical and empirical, more than theoretical -- but overcoming these practical difficulties will be an important part of using RL for economic modeling in the future.

In particular, we do not claim to theoretically guarantee convergence to an equilibrium. Theoretical guarantees around MARL are limited and challenging to find, and proving new ones would be difficult and outside the scope of the paper, especially given the complexity of the simulation environment. Instead, we train agents that empirically produce economically plausible behavior, and empirically test that they at least locally cannot deviate to improve their policies by a large amount (see Table 1 below). To the best of our knowledge, this empirical result is the first of its kind for using RL in such complex economic environments.

| Agent Types | Consumer | Firm | Government |
| ------------------------------ | ---------- | ---------- | ---------- |
| Reward improvement under best response at end of training, as a fraction of the total gain in reward during training| < 0.2% | < 5% | < 0.1% |

Table 1. Reward improvement under best response at the end of training as a fraction of the reward improvement during training, over 10 independent runs (with a few exceptions where the improvements were roughly 3% for consumers, 10% for firms, and 1% for the government).

Update 11/22:

We've added a further updated document, with many improvements suggested by the reviewers or promised in the responses. We also added some additional related work, in particular a paper by Sinitskaya et al. from the agent-based economic modeling literature which presents a model with some similarities to ours, that also finds that learning agents can converge to trivial equilibria where the economy "shuts down". We think qualitatively similar results in a distinct model may further motivate the use of the structured curricula to avoid this outcome.

---

### Decision · Program_Chairs · 2022-01-20

**Decision:**

Reject

**Comment:**

The authors propose a deep multi-agent RL framework to compute equilibria in a economics problem. Several reviewers raised issues with the presentation, as well as issues with evaluating the impact of the work, partly because the novelty of the approach is made insufficiently clear. While the authors have resolved some of the confusions arising from the presentation in their rebuttal, resulting in 2 out of the 4 reviewers to increase their score, the concerns regarding novelty mostly remain. For these reasons, I don’t think this work is ready for publication at ICLR at the moment and recommend rejection.